# JARVIS: TOWARDS PERSONALIZED AI ASSISTANT VIA PERSONAL KV-CACHE RETRIEVAL

## ABSTRACT

The rapid development of Vision-language models (VLMs) enables open-ended perception and reasoning. Recent works have started to investigate how to adapt general-purpose VLMs into personalized assistants. Even commercial models such as ChatGPT now support model personalization by incorporating user-specific information. However, existing methods either learn a set of concept tokens or train a VLM to utilize user-specific information. However, both pipelines struggle to generate accurate answers as personalized assistants. We introduce **Jarvis**, an innovative framework for a personalized AI assistant through personal KV-Cache retrieval, which stores user-specific information in the KV-Caches of both textual and visual tokens. The textual tokens are created by summarizing user information into metadata, while the visual tokens are produced by extracting distinct image patches from the user's images. When answering a question, Jarvis first retrieves related KV-Caches from personal storage and uses them to ensure accuracy in responses. We also introduce a fine-grained benchmark built with the same distinct image patch mining pipeline, emphasizing accurate question answering based on fine-grained user-specific information. Jarvis is capable of providing more accurate responses, particularly when they depend on specific local details. Jarvis achieves state-of-the-art results in both visual question answering and text-only tasks across multiple datasets, indicating a practical path toward personalized AI assistants. The code and dataset will be released.

## 1 INTRODUCTION

Large vision language models (VLMs) have made rapid progress in open ended visual perception and language understanding (Chen et al., 2024; Li et al., 2024a; Bai et al., 2025a; DeepSeek-AI et al., 2024). However, reliably generating accurate answers conditioned on user specific information remains challenging. For example, these models still struggle to consistently recognize the same pet across different photos and to answer questions that depend on fine grained, user specific context. Two failure modes are particularly common: (i) the model often attends to spurious background cues instead of user specific details, and (ii) many existing methods rely on very long prompts, which increase token budgets and introduce latency, instability, and interference between instructions. Together, these issues lead to unreliable identity grounding and limit deployment in interactive, real time applications. These challenges highlight a growing need for personalized VLMs that can robustly maintain user specific identity and context across diverse inputs.

Existing personalization methods can be organized along two orthogonal axes. The first axis addresses whether we update model parameters per concept or keep the backbone fixed; the second axis pertains to where concept information is stored at inference time, either in the prompt, within learned tokens or adapters, or in an external cache. Along the parameter updating axis, token- and adapter based approaches learn lightweight, subject specific parameters from a few images to steer a frozen backbone (Nguyen et al., 2024; 2025; An et al., 2025a). These methods demonstrate that compact concept tokens can capture rich identity information and transfer across tasks. However, they still require per concept optimization and persistent storage of user specific adapters. A complementary line of work keeps the backbone fixed and reorganizes or synthesizes conceptual evidence. This is achieved, for example, by generating structured hierarchies from seed concepts (An et al., 2025b), amortizing personalization via multi concept instruction tuning or reinforcement learning (An et al., 2024; Oh et al., 2025), or mapping reference images into personalized embeddings for a frozen LVLM (Pham

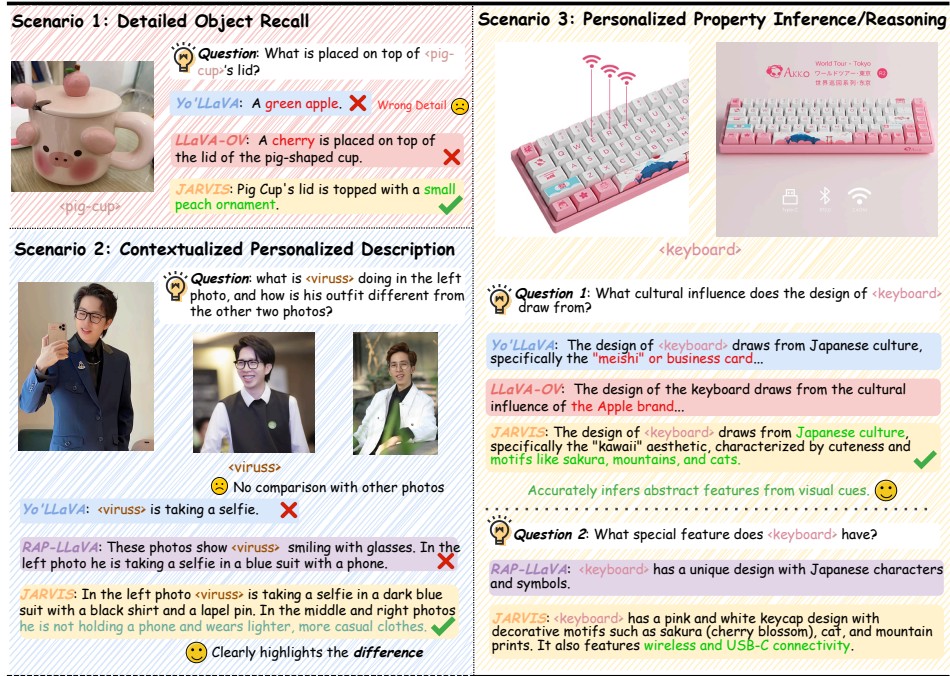

Figure 1: **Qualitative comparison with Yo'LLaVA and RAP-LLaVA across three personalized scenarios.** (**1**) *Detailed object recall:* **Jarvis** correctly recovers fine-grained, instance-specific details (e.g., the small peach ornament) where baselines hallucinate. (**2**) *Contextualized personalized description:* Given multiple reference photos of the same person, **Jarvis** explicitly compares the left photo with the other two images and articulates concrete differences in both action and outfit, whereas baselines either ignore the requested comparison or only give a partial caption of the left image. (**3**) *Personalized property inference/reasoning:* From visual cues, **Jarvis** infers abstract cultural aesthetics and functional features (e.g., "kawaii" motifs) with higher faithfulness and completeness.

et al., 2024). Retrieval centric systems push externalization further by storing user knowledge in key–value memories and conditioning on retrieved exemplars at generation time, thereby reducing maintenance and supporting real time updates (Hao et al., 2025; Das et al., 2025). Despite this progress, current systems often depend on long, concept heavy prompts or large exemplar caches and remain fragile under cluttered scenes, occlusions, and fine grained identity confusions.

A training-free design enables us to prioritize a seamless user experience. We introduce **Jarvis**, which converts conceptual evidence into reusable key–value (KV) states and reuses them across turns without modifying the base model parameters. Qualitative examples in Figure 1 illustrate Jarvis's advantages over Yo'LLaVA and RAP-LLaVA in three personalized visual question answering scenarios. For each concept, the system builds a concise text profile and, in parallel, extracts discriminative visual patches. At inference time, the user query is matched against the indexed concept metadata and patch embeddings, and only the most relevant evidence is attached as external KV instead of concatenating all available evidence into the prompt. Prefetching once and attaching on demand, rather than rebuilding a long context at every turn, shortens prompts, reduces latency and computation, and keeps answers grounded in retrieved regions. We instantiate Jarvis on LLaVA-OneVision (Li et al., 2024a) for text QA, visual QA, and recognition, and design a patch guided evaluation protocol that emphasizes attribute level grounding while keeping all base parameters frozen.

In summary, our contributions are as follows:

- We present Jarvis, a training free personalization framework that inserts concept evidence via external KV prefill and answers queries in a single decoding pass without updating base parameters. We instantiate it on LLaVA-OneVision for both text and visual QA.

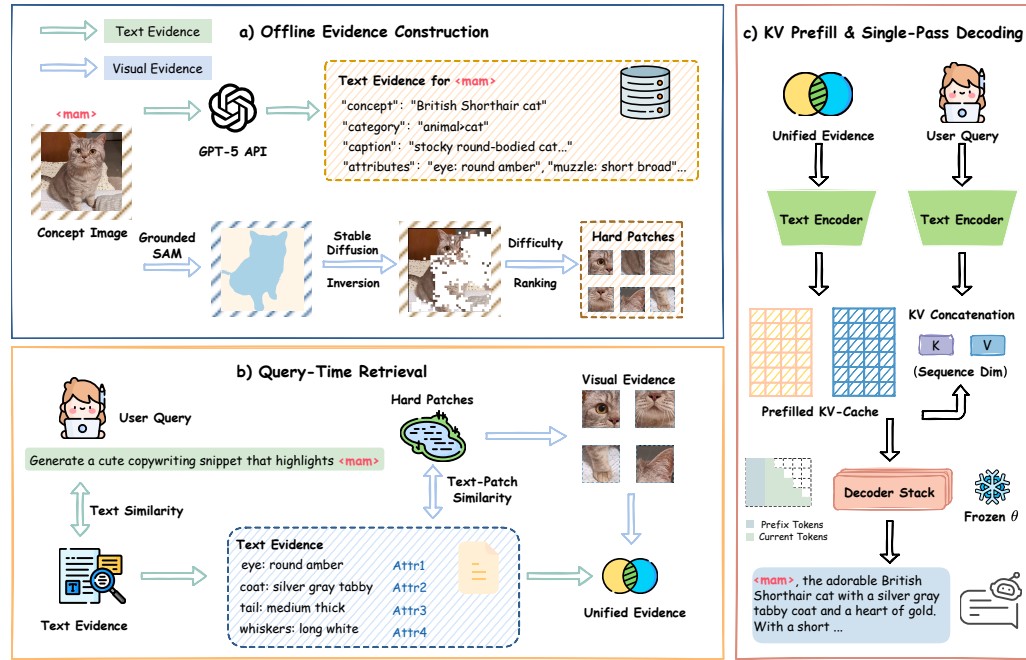

Figure 2: **Model overview.** (**a**) Offline evidence construction: text metadata synthesis and concept-only hard patch mining. (**b**) Query-time retrieval: similarity search over text and visual evidence. (**c**) KV prefill & single-pass decoding: precompute concept KV states and reuse them during decoding.

- We propose evidence-as-KV caching with a unified text–vision pipeline that compiles compact concept metadata and mines highly discriminative hard patches. Retrieved evidence is attached as an external KV rather than concatenated prompts, which reduces context length and latency while preserving faithful grounding.

- We release an open source system and a patch centric dataset. The package includes an end to end implementation (segmentation, hard patch mining, indexing, KV injection, and direct scoring), along with a patch guided QA benchmark augmenting Yo'LLaVA and MC-LLaVA.

## 2 RELATED WORK

**Training-free and retrieval-centric personalization.**   This line of work externalizes user evidence and injects it only at inference time. RAP (Hao et al., 2025) proposes a three-stage pipeline with multimodal retrieval, key-value style memory storage, and on-the-fly conditioning, which simplifies maintenance and supports real-time updates. R2P (Das et al., 2025) follows a training-free paradigm that retrieves discriminative attributes, or concept fingerprints, and reasons over them without any weight edits. These approaches are closely related to multimodal retrieval-augmented generation, which grounds model outputs in retrieved evidence (Mei et al., 2025) and connects to broader work on cross-modal querying and long-term user memory (Abootorabi et al., 2025), subpopulation discovery using large language models (Luo et al., 2024), and agent memory that persists user state beyond the prompt (Packer et al., 2023). A practical bottleneck, however, is that many systems rebuild long prompts at each turn, which increases token usage and latency (Marino et al., 2019). A second limitation is granularity: retrieval often surfaces coarse descriptions or global user profiles instead of attribute-level or region-level cues. This makes it harder to remain faithful in the presence of distractors and compositional attributes. Our work addresses both issues with a training-free alternative that precomputes concept-specific external KV caches and reuses them across turns. These caches are populated with fine-grained text attributes and mined visual patches, which preserve grounding and improve specificity while reducing effective context length and inference costs.

# 3 METHOD

We propose **Jarvis**, a training-free personalization pipeline that injects concept-specific evidence by precomputing and reusing an external key-value (KV) cache. The workflow has three stages aligned with the panels in Fig. 2: (a) offline evidence construction, (b) query-time retrieval, and (c) KV prefill with single-pass decoding. All base-model parameters remain frozen throughout.

## 3.1 PROBLEM DEFINITION

We study session-level personalization for a single, user-specific concept $c \in \mathcal{C}$ within a dialogue turn. A concept $c$ denotes a recurring entity or theme (e.g., a person, pet, or product) that the system resolves at the start of each turn via lightweight retrieval and conditions decoding on the resolved concept to ensure stable grounding and disambiguation across similar contexts.

For each concept we maintain two compact evidence repositories attachable as external key–value (KV) caches rather than prompt tokens: descriptors $T^{(c)}$ and visual/multimodal patches $P^{(c)}$. These repositories are curated offline to be small, highly discriminative, and reusable across turns, thereby enabling lower latency and a shorter practical context while preserving fidelity.

$$T^{(c)} = \left\{t_i^{(c)}\right\}_{i=1}^{m_c}, \qquad P^{(c)} = \left\{p_j^{(c)}\right\}_{j=1}^{n_c}. \tag{1}$$

We instantiate the global repository $\mathcal{R} = \{(T^{(c)}, P^{(c)})\}_{c \in \mathcal{C}}$ via a compact offline evidence-construction pipeline (Section 3.2). Our objective is to produce a response $y$ that is specific to $c$, reliably faithful to the query $q$ (and the image $I$ when present), and robust against closely visually or semantically similar distractor concepts. Formally, given the resolved concept $c$,

$$y = \arg\max_{y'} \Pr\left(y' \,\Big|\, q,\, I,\, T^{(c)},\, P^{(c)},\, \theta\right). \tag{2}$$

At the beginning of each turn, the system resolves the active concept from the user's explicit mention or through lightweight retrieval-based lookup. Then it selects, attaches, and caches the most relevant evidence as external KV for the current session before decoding the final response.

## 3.2 OFFLINE EVIDENCE CONSTRUCTION (FIG. 2A)

**Text metadata.** We employ the multimodal large language model GPT-5 via the official API (OpenAI, 2025) to synthesize a compact textual profile for each concept. Given several representative images and targeted instructions, the model produces a structured record $\mathcal{T}^{(c)}$ comprising four fields: (i) a canonical name; (ii) a category selected from a fixed taxonomy (e.g., animal, person, device); (iii) an approximately 25-word caption summarizing geometry and salient appearance; and (iv) a list of fingerprint attributes formatted as "part: descriptor" (e.g., eye: round amber). We enforce deterministic decoding (temperature = 0, top-p/top-k disabled) and prescribe a strict key-value JSON schema to ensure format compliance. A lightweight postprocessing step standardizes capitalization and tense, enforces the target schema, and removes invalid or ill-formed entries. The resulting $\mathcal{T}^{(c)}$ is compact, cross-concept consistent, and directly usable for retrieval and KV prefill.

**Concept-only hard patches.** We aim to build a small set of patches that summarize the most discriminative visual evidence for each concept across its images, as illustrated in Fig. 3. Instead of mining the entire image, we first localize the subject using GroundingDINO+SAM and restrict candidates to this mask, ensuring that patches do not depend on background shortcuts or incidental scene context (Lin et al., 2025). Inside the mask, we combine two complementary priors: a diffusion–inversion difficulty map that highlights locations where the generative prior struggles to reconstruct, typically coinciding with identity-carrying details, and an OpenCLIP-based text relevance map that focuses on pixels consistent with the concept prompt while suppressing generic co-occurring backgrounds. These signals are fused into a single score map, from which we extract a fixed grid of candidate windows and select a global top-$k$ set of high-scoring patches; each selected patch is encoded using a CLIP image encoder and stored in a visual index, forming the Hard Patch Pool used at retrieval time. A complete step-by-step description of this pipeline, along with design motivations, exact formulas, and hyperparameter values, is provided in Appendix B.

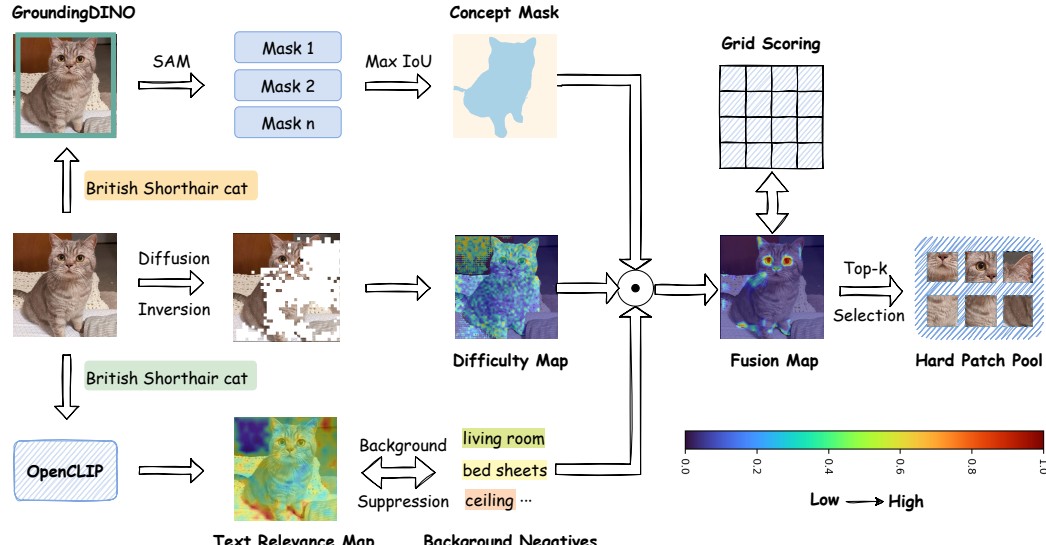

Figure 3: **Concept-only hard patch mining.** We localize the concept, fuse difficulty and text–relevance cues inside the mask, then grid-score to extract top-$k$ informative crops, which populate the visual-indexed *Hard Patch Pool* used by the retriever.

## 3.3 MATERIALIZING EVIDENCE AS EXTERNAL KV (FIG. 2C)

To avoid repeatedly stitching long evidence into the input prompt, we convert each concept's text bundle into a short prefix and run a one-time prefill on the frozen base model $f_\theta$. This produces layer-wise, concept-specific key and value (KV) states that are independent of any particular query and can be stored externally as reusable evidence units. Since the parameters $\theta$ remain frozen, the cost of this prefill is incurred once per concept and is then amortized across many turns and sessions. A detailed description of how we construct these prefixes and caches is provided in Appendix D.

## 3.4 QUERY-TIME RETRIEVAL AND ANSWERING (FIG. 2B,C)

At inference time, the system retrieves a small set of relevant concepts from a joint index built over textual bundles and concept-only hard patches, conditioned on the current query and, when available, the input image. For the retrieved concepts, we gather their cached KV states, assemble them in a deterministic order, and concatenate them along the sequence dimension to form an external prefix. This external prefix is placed in front of the current input's cache, enabling decoding to proceed in a single pass while treating the external KV as additional context. The position indices of the current tokens are offset by the total length of the external prefix, which maintains consistency in relative or rotary position encodings after concatenation. We do not introduce head-wise mixing or learnable gating; the assembly rule is fixed for reproducibility. When the retrieved concept set changes across turns, we prefill only the newly needed concepts and extend the external cache, thereby avoiding repeated construction of long prompts and effectively reducing the context length required for each query. Further implementation details are provided in Appendix D.

## 4 EXPERIMENTS

### 4.1 SETUP

**Tasks and metrics.** We assess concept personalization under three settings: (i) text-only QA, where the model answers questions about a named concept without an image; (ii) visual QA (VQA), where questions refer to a held-out image of the concept; and (iii) recognition, where the model is provided with an image and a target concept name and must determine whether the concept is present in the image. Following the Yo'LLaVA evaluation setup, we adopt accuracy-based metrics for all tasks.

For text-only QA and VQA, we report standard accuracy. For recognition, due to the imbalanced nature of the test set comprising positive and negative examples, we report a weighted accuracy that assigns equal importance to both: $\text{WeightedAcc} = 0.5 \cdot \text{Acc}_{\text{pos}} + 0.5 \cdot \text{Acc}_{\text{neg}}$, where $\text{Acc}_{\text{pos}}$ and $\text{Acc}_{\text{neg}}$ denote the accuracy for images that contain and do not contain the concept, respectively. In all experiments, the backbone remains frozen.

**Compared methods.** All LLaVA-1.5–derived pipelines are re-implemented on the same LLaVA-OneVision (LLaVA-OV) backbone and vision processor, with shared decoding hyperparameters, prompt templates, and evaluation scripts. We report results on the Yo'LLaVA and MC-LLaVA test sets, as well as on our fine-grained ++ variants when applicable.

- **LLaVA-OV+Prompt**: a training-free baseline for LLaVA-OV. We concatenate the query with metadata and concept images into a multi-image context without caching or retrieval.
- **Yo'LLaVA** (Nguyen et al., 2024): single-concept personalization on top of LLaVA. When ported to LLaVA-OV, we reproduce the paper's single-concept protocol.
- **MC-LLaVA** (An et al., 2024): multi-concept personalization. For a strict apples-to-apples comparison, we evaluate only the single-concept slice following the authors' protocol.
- **RAP-LLaVA** (Hao et al., 2025): retrieval-augmented personalization using a concept memory that stores images and attributes, injecting retrieved exemplars at inference time. We evaluate our LLaVA-OV instantiation, consistent with the authors' settings.
- **RePIC** (Oh et al., 2025): an RL-based post-training method for personalizing multi-modal language models by optimizing verifiable rewards for recognition, localization, and identity consistency. We adapt their reinforcement learning pipeline to LLaVA-OV and train using the same reward design and data setup as in the original paper.
- **PLVM** (Pham et al., 2024): an encoder-based personalization approach that uses a frozen DINO-v2 Aligner to map reference images into personalized word and context embeddings for LLaVA. We instantiate the Aligner on top of LLaVA-OV.

**Datasets.** We evaluate the personalization benchmarks **Yo'LLaVA** (Nguyen et al., 2024) and **MC-LLaVA** (An et al., 2024), each organized into episodes with disjoint evidence and evaluation images (no cross-image leakage). In addition to these, we develop **Yo'LLaVA++** and **MC-LLaVA++**: fine-grained, text-only variants constructed from patch-centric evidence (hard patch mining) and GPT-5-based question generation with light human filtering (OpenAI, 2025), explicitly targeting attribute-level grounding and robustness against distractors. The construction pipeline for the ++ datasets is detailed in Appendix E, and summary statistics, sampling details, and qualitative examples are provided in Appendix C.

**Protocol overview.** For each concept, we construct a compact text profile from a small set of evidence images and mine candidate hard patches. We then precompute a concept-specific external KV cache and index the patches. At inference, we score the user query against the concept attributes and attach only the top-matching textual and visual evidence as external KV. In multi-turn sessions on the same concept, cached `past_key_values` sessions are reused to avoid re-prefilling. Unless an ablation specifies otherwise, hyperparameters are shared across datasets.

### 4.2 MAIN RESULTS

Table 1 reports accuracy on *VQA*, *text-only QA*, and *recognition* (Rec) over the original Yo'LLaVA and MC-LLaVA benchmarks and their fine-grained ++ variants. Training-free methods all share the same LLaVA-OV backbone, while Yo'LLaVA, MC-LLaVA, RePIC, and PLVM are trained methods that adapt this backbone with additional parameters. Across the board, **Jarvis** delivers the strongest overall performance: it achieves the best score in seven out of eight columns and is a close second on the remaining MC-LLaVA recognition column, despite keeping the base model frozen.

**Fine-grained sensitivity.** The advantage of Jarvis is most pronounced on the ++ splits that focus on localized, identity-bearing details. On Yo'LLaVA++ and MC-LLaVA++, Jarvis substantially outperforms both training-free and training-required baselines in text-only accuracy, with margins on the order of ten to fifteen points over the strongest competing method. These splits suppress

Table 1: Model comparison on Yo'LLaVA, MC-LLaVA and their ++ variants. All methods, including retrained RAP-LLaVA and other training-required baselines, share the same LLaVA-OneVision (LLaVA-OV) backbone. Methods are grouped into training-free and training-required, and the best and second best performances are highlighted.

| Method | Yo'LLaVA dataset | | | MC-LLaVA dataset | | | Yo'LLaVA++ Text-only | MC-LLaVA++ Text-only |
|---|---|---|---|---|---|---|---|---|
| | VQA | Text-only | Rec | VQA | Text-only | Rec | | |
| *Training-free methods* | | | | | | | | |
| LLaVA-OV+Prompt | 0.959 | 0.823 | 0.960 | 0.937 | 0.812 | 0.831 | 0.702 | 0.679 |
| RAP-LLaVA | 0.917 | 0.795 | 0.943 | 0.844 | 0.828 | 0.947 | 0.625 | 0.592 |
| Ours | 0.970 | 0.865 | 0.967 | 0.941 | 0.871 | 0.935 | 0.856 | 0.835 |
| *Training-required methods* | | | | | | | | |
| Yo'LLaVA | 0.929 | 0.800 | 0.924 | 0.655 | 0.658 | 0.841 | 0.663 | 0.646 |
| MC-LLaVA | 0.934 | 0.800 | 0.947 | 0.844 | 0.710 | 0.878 | 0.629 | 0.636 |
| RePIC | 0.948 | 0.853 | 0.949 | 0.837 | 0.713 | 0.866 | 0.678 | 0.594 |
| PLVM | 0.929 | 0.808 | 0.928 | 0.726 | 0.824 | 0.841 | 0.708 | 0.701 |

background shortcuts and require the model to distinguish visually similar identities, so the gains indicate that combining compact textual fingerprints with mined concept-only hard patches gives the right inductive bias for fine-grained personalization. In contrast, methods that repeatedly concatenate long textual descriptions or multiple images at query time improve the original splits but show much smaller progress on the ++ benchmarks.

**VQA, text-only, and recognition behavior.** Jarvis improves all three metrics simultaneously. On both Yo'LLaVA and MC-LLaVA, Jarvis leads all training-free methods on VQA and text-only accuracy, while achieving the best recognition on Yo'LLaVA and essentially matching the top recognition score on MC-LLaVA. This pattern is consistent with the design: converting concept metadata and hard patches into external key–value states provides a stable semantic prior without extending the token context, which primarily benefits text-only decoding. At the same time, concept-only hard patches keep the visual pathway focused on subject regions rather than backgrounds, preserving strong VQA performance. Training-based methods such as RePIC and PLVM achieve competitive recognition scores but leave noticeable headroom on text-only QA, suggesting that their learned adapters do not fully resolve identity ambiguity when no image is present.

**Training-free vs. training-required.** Perhaps most strikingly, Jarvis as a purely training-free method matches or exceeds the best training-required baselines in nearly every setting. Yo'LLaVA, MC-LLaVA, RePIC, and PLVM do provide gains over the LLaVA-OV+Prompt baseline, especially on recognition and some VQA columns, but they still fall short of the external-KV approach, particularly on the harder ++ splits. This suggests that materializing concept evidence as reusable KV caches, instead of encoding it anew as tokens or parameters, offers a more direct and controllable interface for personalization: it keeps the effective context short, reduces interference across turns, and lets mined hard patches supply the localized visual evidence that fine-grained evaluation stresses the most.

## 4.3 LATENCY AND THROUGHPUT

We measure end to end responsiveness and serving capacity under identical hardware and decoding settings: LLaVA-OV backbone and vision tower, greedy decoding, a fixed maximum token budget, and four concurrent clients. For each concept we vary the number of shared queries per session $Q \in \{1, 2, 4, 8, 16, 32\}$ and report two metrics: (i) wall to wall average per turn latency in milliseconds with explicit CUDA synchronization, and (ii) throughput in queries per second (QPS), defined as completed requests divided by elapsed wall time at the fixed concurrency level. Unless otherwise noted, the user query is *Tell me <mam>'s ear shape, eye color, and hair length*. We evaluate seven personalization pipelines on the same evidence and prompt templates: Prompt-concat, **Jarvis**, Yo'LLaVA, RAP-LLaVA, MC-LLaVA, PLVM, and RePIC.

Figure 4 shows the results. Across all values of $Q$, **Jarvis** achieves the highest QPS and the lowest latency by a clear margin. Jarvis performs a single prefill per concept to construct an external key–value (KV) cache that already encodes the concept text and its mined hard patches. At inference

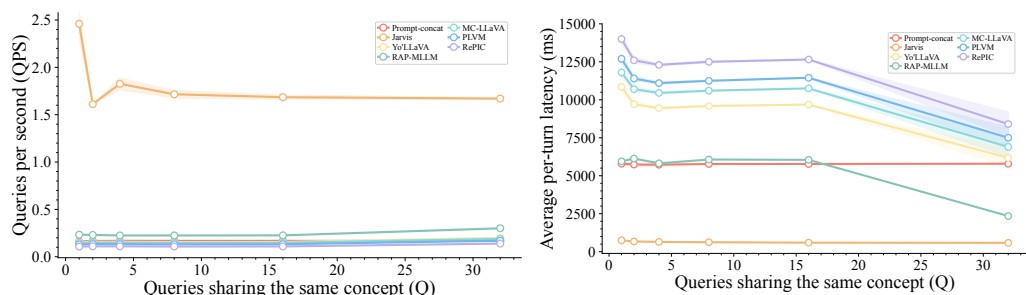

Figure 4: Throughput and latency under personalization. (Left) QPS vs. $Q$. (Right) average per-turn latency vs. $Q$. All methods share hardware/decoding, meaning that 95% CIs are consistent across trials. Higher is better for QPS; lower is better for latency.

time, we only attach a short list of retrieved attributes and patch keys, then decode the user query in one pass on a compact context. As $Q$ grows, the fixed prefill cost is amortized over many turns, so per turn latency stays near the sub second regime while QPS increases steadily.

The remaining methods all keep concept evidence in token form and therefore rebuild longer prompts at every turn. Prompt-concat is the most expensive configuration: it concatenates the query with text metadata and all concept images each time, which produces the lowest QPS (around 0.15 to 0.18) and the highest latency. Yo'LLaVA, MC-LLaVA, PLVM, and RePIC personalize the model through additional tokens or adapters but still rely on multi image prompts during inference; they cluster around similar throughput and latency and do not benefit much from larger $Q$. RAP-LLaVA retrieves concept exemplars dynamically and is faster than Prompt-concat, yet it still injects retrieved evidence as regular tokens and therefore remains far behind Jarvis in both QPS and latency.

**Takeaways.** (1) A reusable, prefilled KV cache turns per turn prompt construction into a one time setup. This reduces the effective context length for every subsequent query and yields both lower latency and higher throughput as $Q$ increases. (2) Attaching only top $k$ retrieved attributes together with a small number of mined hard patches keeps the external KV compact and focused, which leads to stable decoding behavior even under high concurrency. (3) For repeated queries about the same concept, Jarvis consistently delivers lower user perceived latency and higher server side efficiency than token based personalization schemes such as Prompt-concat, Yo'LLaVA, MC-LLaVA, PLVM, RePIC, and retrieval systems that do not operate directly in KV space.

### 4.4 Ablation Studies

We analyze **Jarvis** from two angles. First, we study how retrieval and indexing hyperparameters affect performance and show that the method is stable under a wide range of settings (Fig. 5). Second, we turn individual evidence channels on and off to understand where the gains come from (Tab. 2).

#### 4.4.1 Hyperparameter ablation.

We first probe how **Jarvis** behaves as we vary the main retrieval knobs. Fig. 5(a) sweeps the number of retrieved textual attributes $k_{\text{attr}}$ and retrieved visual patches $k_{\text{vis}}$ at test time. Across all four benchmarks, text-only accuracy consistently improves from $k_{\text{attr}}{=}2$ to $k_{\text{attr}}{=}4$, but then saturates or slightly declines at $k_{\text{attr}}{=}8$. This pattern suggests that the model benefits from a compact, well-focused attribute set that pins down the concept, while longer lists start to introduce redundancy and mild semantic drift. A similar trend appears for $k_{\text{vis}}$: moving from 2 to 4 patches brings stable gains, whereas $k_{\text{vis}}{=}8$ yields only marginal improvements. Together, these trends indicate that Jarvis operates in a regime where a few strong signals are more valuable than an exhaustive description, and where retrieval hyperparameters admit a broad "good" region rather than a brittle optimum. We therefore adopt $k_{\text{attr}}{=}4$ and $k_{\text{vis}}{=}4$ as a simple, robust default.

Fig. 5(b) varies the number of mined hard patches per image $k_{\text{patch}}$ used when building the visual index. Accuracy improves monotonically from $k_{\text{patch}}{=}1$ to $k_{\text{patch}}{=}4$ and then essentially plateaus at

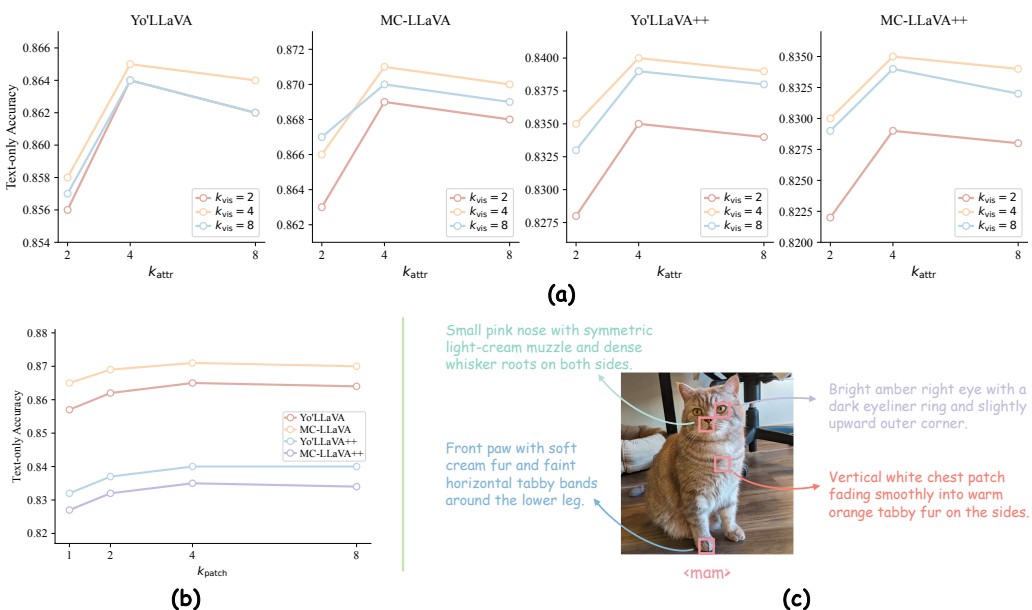

Figure 5: Ablations of hard-patch-guided retrieval and indexing. (a) Effect of the number of retrieved attributes $k_{\text{attr}}$ and retrieved visual patches $k_{\text{vis}}$ on text-only accuracy across all four benchmarks. (b) Effect of the number of mined hard patches per image $k_{\text{patch}}$ when building the visual index. (c) Qualitative visualization of concept-only hard patches for concept <mam>, showing the selected regions and their fine-grained textual descriptions.

$k_{\text{patch}}=8$. This suggests that a small handful of patches per image is already sufficient to capture the key identity-bearing regions, and that adding more patches mostly recycles similar evidence. We therefore set $k_{\text{patch}}=4$ in all main experiments. Finally, Fig. 5(c) provides a qualitative example for concept <mam>: the selected patches concentrate on the nose, chest, front paw, and right eye, and their associated textual descriptions emphasize distinctive local traits rather than global background. This supports our design intuition that hard-patch mining distills a compact, high-value visual summary that aligns well with the attribute-centric retrieval mechanism used at inference time.

### 4.4.2 COMPONENT-WISE ABLATION.

We next ablate three evidence channels in **Jarvis** by toggling them on or off: *QA-Attr* (textual attributes drawn from the metadata field), *VisPatch* (retrieved hard-patch visual cues), and *BGS* (background suppression that keeps limited local context when forming each hard patch). Results on Yo'LLaVA and MC-LLaVA, together with their fine-grained ++ splits, are summarized in Tab. 2.

**QA-Attr (textual attributes).** Removing QA-Attr produces the largest degradation across all variants in Tab. 2. The drop is most severe for text-only QA and becomes even more pronounced on the ++ splits; VQA performance also suffers. This pattern indicates that textual attributes act as the primary semantic prior for both identity-level and attribute-level personalization. Without this channel, the model behaves much closer to a generic VLM and systematically fails on fine-grained references, even when other signals are still available.

**VisPatch (hard-patch visual evidence).** Disabling VisPatch leaves standard VQA performance almost unchanged but consistently lowers text-only accuracy and degrades results on the ++ splits for both datasets. This aligns with the picture from Fig. 5: localized, hard-to-synthesize visual cues matter most when the task hinges on subtle appearance differences, which are exactly what the fine-grained benchmarks are designed to stress. In contrast, coarse recognition on the original VQA splits can already be handled reasonably well using the textual channel alone.

**BGS (background suppression in patches).** BGS suppresses most of the surrounding background when forming hard patches while preserving a small amount of local context. Turning BGS on yields

Table 2: Ablations of **Jarvis** components on the same datasets/metrics as Table 1. Yo'=Yo'LLaVA, MC=MC-LLaVA, "++" denotes the fine-grained split. VQA denotes VQA accuracy; Txt denotes text-only accuracy. ✓=enabled, ✗=disabled.

| Components | | | Yo' | | MC | | Yo'++ | MC++ |
|---|---|---|---|---|---|---|---|---|
| QA-Attr | VisPatch | BGS | VQA | Txt | VQA | Txt | Txt | Txt |
| ✓ | ✓ | ✓ | **0.970** | **0.865** | **0.941** | **0.871** | **0.856** | **0.835** |
| ✓ | ✓ | ✗ | 0.959 | 0.850 | 0.936 | 0.860 | 0.846 | 0.824 |
| ✓ | ✗ | ✓ | **0.970** | 0.855 | 0.939 | 0.860 | 0.842 | 0.823 |
| ✗ | ✓ | ✓ | 0.935 | 0.703 | 0.924 | 0.662 | 0.627 | 0.573 |

small but consistent gains, including on the ++ splits; turning it off leads to reproducible drops. This suggests that reducing background clutter removes noisy correlations during retrieval and leads to cleaner information being stored in the cached KV states than either fully foreground-only or background-retaining patches.

**Cross-variant observations and takeaway.** Across all datasets, the same ordering emerges: enabling all three channels performs best; removing BGS or VisPatch leads to moderate reductions; removing QA-Attr causes the most drastic collapse. The ++ splits are more sensitive than the standard splits, reflecting their reliance on fine-grained and localized cues. Overall, QA-Attr provides the semantic scaffold, VisPatch injects discriminative visual details, and BGS improves the robustness of what is written into the external memory. Combining all three channels yields a simple and transferable configuration that achieves strong performance without changing any backbone parameters.

## 5 CONCLUSION

We introduce a training-free personalization framework that externalizes concept evidence into reusable KV caches and attaches them as a short decoding prefix. This amortizes context processing across turns, reducing token and latency costs while preserving grounding. Compared with light finetuning, external KV improves time-to-first-answer and serving efficiency without per-user adapters. We also enhance the dataset with fine-grained supervision, including attribute phrases and region-level patches obtained through automatic mining with lightweight human verification, to increase specificity and better stress-test distractors. Looking ahead, key directions include principled cache composition (for example, routing or sparse attention), confidence-aware gating of cache usage, and memory-efficient compression with privacy-preserving storage. Together, these steps retain the throughput gains of **Jarvis** while improving robustness in open-world deployments.

## ETHICS STATEMENT

All experiments in this study were conducted using publicly available datasets and adhered to the corresponding licenses. No new data collection or clinical trials were performed, and no human subjects research requiring additional consent or IRB approval was involved. The work does not introduce privacy or security risks beyond those inherent in the public datasets. We declare no conflicts of interest or sponsorships related to this research.

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

# A    LLM USAGE

Throughout the implementation and debugging process, we consulted large language models for targeted technical guidance. Following the collaborative drafting of the manuscript, we utilized LLMs to refine the wording, enhance clarity, and improve the overall presentation of the text.

# B    DETAILS FOR HARD PATCH MINING

This section provides a complete description of the hard patch mining procedure used to build the visual index of concept-only patches. The goal is to extract a small set of image regions that are both strongly related to the target concept and inherently difficult for the generative prior to reproduce, while avoiding reliance on background context. Algorithm 1 summarizes the overall concept-only hard patch mining pipeline, and the following subsections detail each component.

## B.1    PREPROCESSING AND NOTATION

For each concept, we are provided with a set of training images $\{I_m\}_{m=1}^{M}$ and an associated concept text $T$. Each image $I_m$ is first resized to a fixed resolution of $512 \times 512$ and normalized to the input space of the diffusion model and the vision encoder. For clarity, we summarize the main symbols in Table 3 and use the same notation as in Algorithm 1 of the main paper.

Given an image $I_m$, we denote $M_m \in \{0,1\}^{H \times W}$ as the subject mask that covers the primary instance of the concept within that image. The mask is obtained using an open vocabulary detector conditioned on $T$, followed by a segmentation model that refines the region into a dense binary mask. We retain the largest connected component to focus on the main subject while suppressing small, spurious detections.

To define patch level candidates, we use a collection of square windows indexed by $(u, v)$. For each index $(u, v)$, we denote $B_m(u, v) \subset \{1, \ldots, H\} \times \{1, \ldots, W\}$ as the set of pixel coordinates belonging to that window in image $I_m$. In the implementation, these windows are obtained using a dense sliding procedure across a small set of patch sizes, which is described later in this section. For the theoretical description, it is sufficient to view $B_m(u, v)$ as an abstract patch, with its shape and stride determined by hyperparameters.

## B.2    DIFFICULTY MAP FROM DIFFUSION INVERSION

The first step is to estimate how challenging each spatial location in $I_m$ is for a large text to image diffusion model when the model is not provided with any concept specific guidance. The key idea is to treat the pretrained diffusion model as a strong prior over generic natural images. Regions that fit this prior well are considered easy, while regions that require the model to work harder to match the observations are regarded as difficult and are likely to contain subject specific details.

We use diffusion inversion to obtain a latent trajectory that reconstructs $I_m$. Starting from the observed image $I_m$, we run the diffusion process backward to recover a sequence of latent variables

$$\{z_t^{(m)}\}_{t=1}^{T} \quad \text{with} \quad z_1^{(m)} \approx I_m. \tag{3}$$

We then run the diffusion model forward from $\{z_t^{(m)}\}$ using an unconditional or empty text input and monitor how the latent representation changes over time at each spatial location. If a location corresponds to a typical background region, such as a smooth wall or an untextured piece of cloth, the unconditional model can reconstruct it with minor corrections, and the latent representation quickly stabilizes. If a location corresponds to a fine grained pattern tied to the instance identity, such as the eyes of a particular pet or a rare texture, the unconditional model repeatedly adjusts that location to reconcile the prior with the observation.

From this process, we derive a difficulty map

$$C_m \in [0, 1]^{H \times W}, \tag{4}$$

Table 3: Notation and hyperparameters for hard patch mining.

| Symbol or parameter | Meaning | Typical value |
|---|---|---|
| $I_m$ | Image for a concept | $512 \times 512$ |
| $T$ | Concept text prompt | Dataset prompt |
| $\mathcal{B}$ | Background negative text set | Fixed generic list |
| $M_m$ | Subject mask for $I_m$ | Largest component |
| $C_m$ | Difficulty map from diffusion inversion | $[0,1]^{H \times W}$ |
| $R_m^+$ | Text relevance map for $T$ | Vision language model |
| $R_{m,b}^-$ | Relevance map for background $b$ | Same as $R_m^+$ |
| $\tilde{R}_m$ | Background suppressed relevance map | Normalized difference |
| $C_m^w$ | Fused difficulty relevance inside mask | $C_m \tilde{R}_m^\gamma$ on $M_m$ |
| $\gamma$ | Exponent for text relevance | 1.0 or 1.5 |
| $\mathcal{P}$ | Set of patch sizes | $\{32, 24, 16, 8\}$ |
| $B_m(u,v)$ | Patch support in $I_m$ | Window at a given scale |
| $s_m(u,v)$ | Patch score | Mean of $C_m^w$ on $B_m(u,v)$ |
| $\kappa_m(u,v)$ | Mask coverage of a patch | Used for filtering |
| $\eta$ | Minimum coverage threshold | 1.0 |
| $d$ | Number of mask dilation steps | 0 or 1 |
| $k$ | Maximum patches per image | 4 |

which assigns a scalar difficulty score to each pixel. In practice, we accumulate the magnitude of latent changes along the forward denoising trajectory at each spatial position and normalize the result to the interval $[0,1]$. Pixels that exhibit large or persistent updates are assigned higher values because the unconditional diffusion prior requires many nontrivial corrections to align with them. Pixels that exhibit minimal changes receive low values, as previously explained.

The exact implementation details of this accumulation can vary, but the underlying motivation remains consistent. The pretrained diffusion model captures common structures in natural images. Wherever the model can reconstruct the image in an unconditional setting with minimal effort, we regard the region as easy and unlikely to carry distinctive information about the concept instance. Wherever the model struggles and repeatedly modifies the latent representation, we consider the region to be difficult and potentially rich in concept specific cues.

This leads to the central motivation for using diffusion inversion in hard patch mining. The map $C_m$ is an instance specific prior that highlights locations where the generative prior finds the image surprising or atypical. These locations often contain high frequency details, rare patterns, and identity carrying components that are crucial for personalization; however, they are not determined solely by the text prompt. By combining $C_m$ with text based relevance in later steps, we bias patch mining toward regions that are semantically related to the concept and are intrinsically difficult for the unconditional diffusion model. This combination enables the method to focus on patches that truly characterize the individual concept rather than on generic backgrounds that the model already knows how to synthesize.

### B.3 TEXT RELEVANCE AND BACKGROUND SUPPRESSION

The second step is to estimate how strongly each pixel in $I_m$ is related to the concept text $T$. We employ a vision language model from the CLIP family and construct a dense relevance map through a gradient based localization procedure. For each image and text pair $(I_m, T)$, we obtain

$$R_m^+ \in [0,1]^{H \times W}, \tag{5}$$

where larger values indicate stronger alignment between the local visual features and the concept text.

To avoid mining patches that rely on background context, we introduce a small set of background negative prompts, denoted by $\mathcal{B}$. These prompts correspond to common but non identifying elements that frequently co occur with many subjects, such as *background*, *wall*, *floor*, *ceiling*, *bed*, *blanket*, *sheet*, *pillow*, *furniture*, *table*, and similar words. For each $b \in \mathcal{B}$, we compute a relevance map

$$R_{m,b}^- \in [0,1]^{H \times W} \tag{6}$$

---

**Algorithm 1** Concept-only hard patch mining

---

**Require:** Concept images $\{I_m\}_{m=1}^M$; concept text $T$; background negatives $\mathcal{B}$; grid size $g$; top-$k$; fusion exponent $\gamma$; minimum mask area $\tau$; minimum in-mask coverage $\eta$.
**Ensure:** Hard patch set $\mathcal{P}^{(c)}$ with metadata; visual index $\mathcal{I}$.

1: $\mathcal{P}^{(c)} \leftarrow \emptyset$; $\mathcal{I} \leftarrow \emptyset$; $\mathcal{C}_{\text{cand}} \leftarrow [\,]$      ▷ global candidate list
2: **for** $m \leftarrow 1$ **to** $M$ **do**
3:      **Subject mask:** run GroundingDINO + SAM on $(I_m, T)$; refine with SAM; keep the largest connected component as $M_m$.
4:      **if** $\text{area}(M_m) < \tau$ **then then continue**      ▷ discard images with tiny subjects
5:      **end if**
6:      **Difficulty map:** compute $C_m$ via Stable Diffusion inversion with empty prompt; normalize to $[0,1]^{H \times W}$.
7:      **Text relevance with background suppression:** obtain OpenCLIP relevance $R_m^+$ for $T$ and background maps $\{R_{m,b}^-\}_{b \in \mathcal{B}}$; set

$$R_m \leftarrow \text{normalize}\left(\text{ReLU}\left(R_m^+ - \max_{b \in \mathcal{B}} R_{m,b}^-\right)\right).$$

8:      **Fusion within mask:** $C_m^{\text{w}} \leftarrow \text{normalize}\left(C_m \cdot (R_m)^\gamma\right) \odot M_m$.
9:      **Grid scoring:** tile a fixed $g \times g$ grid on $I_m$.
10:      **for each** cell $(u, v)$ with box $B_m(u, v)$ **do**
11:          $\kappa \leftarrow \dfrac{|B_m(u,v) \cap M_m|}{|B_m(u,v)|}$      ▷ fraction of subject pixels
12:          **if** $\kappa < \eta$ **then then continue**      ▷ skip patches with insufficient subject coverage
13:          **end if**
14:          $s_m(u,v) \leftarrow \text{mean}_{(x,y) \in B_m(u,v)} C_m^{\text{w}}(x,y)$
15:          append $(m, u, v, s_m(u,v))$ to $\mathcal{C}_{\text{cand}}$
16:      **end for**
17: **end for**
18: **Global selection:** sort $\mathcal{C}_{\text{cand}}$ in descending order by $s_m(u,v)$; keep the first $k$ entries.
19: **for each** $(m, u, v, s_m(u,v))$ in the truncated $\mathcal{C}_{\text{cand}}$ **do**
20:      crop patch $p$ from $I_m$ at $B_m(u,v)$; record its box, $s_m(u,v)$, and other statistics.
21:      $f(p) \leftarrow \text{CLIP-image-encoder}(p)$.
22:      insert $(f(p), \text{metadata})$ into the visual index $\mathcal{I}$; add $p$ (with metadata) to $\mathcal{P}^{(c)}$.
23: **end for**
24: **return** $\mathcal{P}^{(c)}, \mathcal{I}$

---

using the same vision language model, but now conditioned on the background text $b$.

We then construct a background suppressed relevance map

$$\tilde{R}_m(x,y) = \max\left\{0, R_m^+(x,y) - \max_{b \in \mathcal{B}} R_{m,b}^-(x,y)\right\}, \tag{7}$$

followed by a normalization step that rescales $\tilde{R}_m$ to the interval $[0, 1]$. A pixel is considered a strong positive location only if its relevance to the concept text $T$ exceeds its relevance to every background prompt. This operation reduces the influence of frequently co occurring backgrounds and encourages the method to focus on concept specific evidence.

### B.4 Fusion with the Subject Mask

The difficulty map $C_m$ and the background suppressed relevance map $\tilde{R}_m$ capture complementary information. The former highlights instance specific regions that are challenging for the unconditional generative prior, while the latter captures semantic alignment with the concept text after removing obvious backgrounds. To focus these signals solely on the subject region, we combine them within the mask $M_m$.

We first introduce a fusion exponent $\gamma > 0$ that controls the relative importance of text relevance. The fused map is then defined as

$$C_m^w(x,y) = \begin{cases} \text{normalize}\big(C_m(x,y) \cdot \tilde{R}_m(x,y)^\gamma\big), & \text{if } M_m(x,y) = 1, \\ 0, & \text{otherwise,} \end{cases} \quad (8)$$

where the normalization again rescales the nonzero values to the interval $[0,1]$. If $\gamma$ is larger than one, the fusion emphasizes regions that are both challenging and text relevant. If $\gamma$ is close to one, the fusion treats both sources of information in a more balanced manner. In all experiments, we utilize a fixed value of $\gamma$ across concepts.

This fused map $C_m^w$ assigns high scores to pixels that lie within the subject mask, are semantically aligned with the concept text, and are difficult to reconstruct without that text. It therefore serves as the primary signal used for mining concepts related solely to hard patches.

## B.5 MULTI SCALE PATCH PROPOSALS AND COVERAGE CONSTRAINT

To convert the fused map $C_m^w$ into patch candidates, we consider a set of square patch sizes

$$\mathcal{P} = \{p_1, p_2, \ldots, p_S\}, \quad (9)$$

expressed in pixels on the rescaled $512 \times 512$ grid. In practice, we start with a list of user selected patch sizes, such as $32, 24, 16, 8$, and sort them from large to small. The mining procedure employs a dense sliding window for each $p \in \mathcal{P}$, with a stride proportional to the patch size. For each scale $p$, multiple overlapping windows are generated, each defining a candidate support $B_m(u,v)$.

For each candidate window, we compute two quantities. First, we evaluate its fused difficulty score by averaging $C_m^w$ over the window

$$s_m(u,v) = \frac{1}{|B_m(u,v)|} \sum_{(x,y) \in B_m(u,v)} C_m^w(x,y). \quad (10)$$

This quantity serves as the primary measure of patch quality. Second, we measure the fraction of the patch that is covered by the subject mask

$$\kappa_m(u,v) = \frac{|B_m(u,v) \cap M_m|}{|B_m(u,v)|}. \quad (11)$$

We introduce a coverage threshold $\eta \in (0,1]$ and require $\kappa_m(u,v) \geq \eta$. In our default setting, we use $\eta = 1.0$, which ensures that every retained patch lies entirely within the subject mask and contains no background pixels.

The coverage test is efficiently implemented using an integral image of $M_m$. This enables the evaluation of the number of foreground pixels within any window $B_m(u,v)$ in constant time. We first collect patches using the largest patch size in $\mathcal{P}$. If we cannot gather enough patches that satisfy the coverage constraint, we proceed to the next smaller patch size and continue. This top down strategy prefers larger patches whenever possible but reverts to smaller patches when the subject occupies a limited area.

## B.6 MASK DILATION AS FALLBACK

In some cases, the subject mask is slightly conservative; for example, the segmentation model may leave a small gap around fine structures, such as hair or thin limbs. In these situations, the strict coverage constraint can make it difficult to find enough patches, especially at larger scales. To address this, we provide an optional fallback that applies a small number of binary dilation steps to the mask.

Let $d$ denote the number of dilation steps. We construct a dilated mask $\tilde{M}_m$ by repeatedly expanding the foreground region using a $3 \times 3$ neighborhood. If the initial mining pass with $M_m$ does not return the desired number of patches, and $d > 0$, we repeat the procedure using $\tilde{M}_m$ in place of $M_m$ and

recompute the fused map $C_m^w$ accordingly. This method is more robust to small under segmentation errors while still respecting the intended subject region.

If the dilated mask cannot provide enough patches that satisfy the coverage constraint, the mining process reports a failure for that image. In practice, this rarely occurs when the subject occupies a nontrivial fraction of the frame.

### B.7 SELECTION AND VISUAL INDEX CONSTRUCTION

For each image, we aim to extract at most $k$ hard patches, where $k$ is a small integer, such as $4$. During mining, we first generate many more candidate windows than needed, on the order of $50k$ per image, by sweeping the fused map across all patch sizes in $\mathcal{P}$. From this pool, we retain at most $k$ candidates that satisfy the coverage constraint and maximize the score $s_m(u, v)$.

Each retained patch is cropped from the rescaled image and resized to match the input resolution of the vision encoder. We embed the patch into a feature vector using a pretrained image encoder and add this vector to the visual index, along with metadata that records the concept key, the original image path, the patch box $B_m(u, v)$ in image coordinates, the score $s_m(u, v)$, and auxiliary statistics summarizing how local difficulty accumulates along the diffusion trajectory. The encoder and index are shared across all concepts.

The result is a compact, concept conditioned visual index that stores a small number of concept only hard patches for each subject. At inference time, this index can be used to retrieve informative visual evidence for the concept without relying on any background patterns.

## C DATASET STATISTICS

We benchmark on **Yo'LLaVA** and **MC-LLaVA**, two personalization suites organized as concept-centric episodes with disjoint evidence and evaluation images (Nguyen et al., 2024; An et al., 2024). To emphasize fine-grained reasoning without additional training, we also construct text-only variants **Yo'LLaVA++** and **MC-LLaVA++** by guiding GPT-5 with mined hard patches and light human filtering (OpenAI, 2025). Here, "Evid./concept" denotes the number of images used solely to build evidence; evaluation images never overlap with evidence. Following the `yollava-data` protocol, we report MC-LLaVA on its single-concept split for comparability. Summary statistics are presented in Table 4.

Table 4: Datasets used for evaluation. "Evid./concept" is the number of images used to construct evidence; evaluation images are disjoint. Yo'LLaVA++ and MC-LLaVA++ are fine-grained variants created via hard-patch guidance and GPT-5 generation with human filtering.

| Dataset | Modality | # Concepts | Evid./Concept | # QA Pairs | Avg. Q Len. | Split |
|---|---|---|---|---|---|---|
| Yo'LLaVA | Text & Visual | 40 | 5 | 570 | 6.03 | test |
| MC-LLaVA | Text & Visual | 118 | 5 | 1055 | 7.16 | test |
| Yo'LLaVA++ | Text-only | 40 | 5 | 480 | 7.44 | test |
| MC-LLaVA++ | Text-only | 118 | 5 | 1416 | 8.07 | test |

## D EXTERNAL KV CONSTRUCTION AND DECODING

### D.1 EXTERNAL KV CONSTRUCTION

For each personalization concept $c$ we maintain a compact text bundle

$$T^{(c)} = \{\text{concept, category, caption, fingerprint\_attributes}\}. \tag{12}$$

In practice this bundle is obtained by the Four-Key Metadata Prompt and stored as a small JSON object.

To obtain the short prefix $\tau^{(c)}$ used for external KV construction, we apply a deterministic linearization procedure:

1. **Attribute filtering.** We start from the list `fingerprint_attributes` and discard entries that only describe background or context (for example "often: on a bed", "often: in a living room") or pure logos. We keep at most $K$ subject-centric attributes (typically $K \le 16$) that describe stable local traits of the subject.

2. **Template-based linearization.** The remaining fields in $T^{(c)}$ are concatenated into a single natural-language prefix using a fixed template, for example

$$\tau_{\text{text}}^{(c)} = \text{"Subject: } \texttt{concept}. \text{ Category: } \texttt{category}. \\ \text{Description: } \texttt{caption}. \text{ Distinctive traits: } \texttt{attr\_1; ...; attr\_K}." \tag{13}$$

where `attr_k` are short telegraphic phrases taken from the filtered fingerprint attributes. This mapping from $T^{(c)}$ to the string $\tau_{\text{text}}^{(c)}$ is fully rule-based and does not involve any trainable parameters.

3. **Tokenization and truncation.** We tokenize $\tau_{\text{text}}^{(c)}$ with the tokenizer of the frozen base model $f_\theta$ and truncate the resulting token sequence to a maximum length $L_{\max}$ (on the order of a few dozen tokens). The final sequence of tokens is denoted by $\tau^{(c)}$.

Once $\tau^{(c)}$ is obtained, we run a one-time prefill on the frozen base model $f_\theta$ to obtain layer-wise key–value states:

$$\left(\mathbf{K}_{1:L}^{(c)}, \mathbf{V}_{1:L}^{(c)}\right) = \text{Prefill}\left(f_\theta, \tau^{(c)}\right). \tag{14}$$

We store these in an external cache

$$\text{KV}^{(c)} = \{(\mathbf{K}_\ell^{(c)}, \mathbf{V}_\ell^{(c)})\}_{\ell=1}^{L}, \tag{15}$$

and reuse them across turns; the model parameters $\theta$ remain frozen throughout.

## D.2 RETRIEVAL AND ASSEMBLY

In the inference implementation, text and vision are retrieved separately. The small concept set $\mathcal{S}(q, I)$ is determined purely from the textual query $q$; the image $I$ is only used later by the underlying vision–language model when answering the question.

**Offline semantic index (text only).** From each bundle $T^{(c)}$ we build a short semantic document

$$D^{(c)} = \text{concept} + \text{caption} + \text{top-}K \text{ fingerprint\_attributes}, \tag{16}$$

where "+" denotes simple string concatenation after cleaning and truncation. We then encode this document with the base model's token embedding layer and mean-pool across tokens:

$$\mathbf{e}_c^{\text{sem}} = \text{LMEmb}\left(D^{(c)}\right), \tag{17}$$

followed by $\ell_2$-normalization. This yields a single text-only semantic vector per concept.

**Online query encoding and concept retrieval.** Given a user query $q$, we encode it in exactly the same way:

$$\mathbf{e}_q^{\text{sem}} = \text{LMEmb}(q), \tag{18}$$

again normalized to unit length. For each concept $c$ we compute a cosine similarity score

$$s_{\text{text}}(c \mid q) = \left\langle \mathbf{e}_q^{\text{sem}}, \mathbf{e}_c^{\text{sem}} \right\rangle. \tag{19}$$

If the query explicitly mentions placeholder identifiers such as $\langle \texttt{cat-cup} \rangle$, we directly map these placeholders to the corresponding concept IDs and use them as $\mathcal{S}(q, I)$. Otherwise, we select the top-$m$ concepts based on text similarity:

$$\mathcal{S}(q, I) = \{c_1, \ldots, c_m\} = \text{Top-}m_c \, s_{\text{text}}(c \mid q). \tag{20}$$

Here $m$ is a small constant (e.g., $m \le 2$ in our experiments) that ensures the external prefix remains compact.

**Optional visual evidence (separate from retrieval).** Independently of the above text-based selection, we maintain an optional CLIP-based index over hard patches for each concept. Given $q$, we may use CLIP text embeddings of $q$ to retrieve a few representative patches per selected concept and convert them into short "[VIS]" textual hints (for example, ""the left ear region shows dark, edged fur"). These hints are appended to the evidence text but do not change $\mathcal{S}(q, I)$ or the retrieval scores.

**External cache assembly.** Given $\mathcal{S}(q, I)$, we concatenate the per-concept caches to form a single external prefix. Let the total external prefix length be

$$L_{\text{ext}} = \sum_{c \in \mathcal{S}(q,I)} \left| \tau^{(c)} \right|. \tag{21}$$

We assemble per-layer external caches by sequence-wise concatenation:

$$\mathbf{K}_\ell^{\text{ext}} = \text{concatseq}\left(\mathbf{K}_\ell^{(c_1)}, \ldots, \mathbf{K}_\ell^{(c_m)}\right), \tag{22}$$

$$\mathbf{V}_\ell^{\text{ext}} = \text{concatseq}\left(\mathbf{V}_\ell^{(c_1)}, \ldots, \mathbf{V}_\ell^{(c_m)}\right). \tag{23}$$

## D.3   ONE-PASS DECODING

Given the current input $(q, I)$, the base model produces its own per-layer key–value states for the current tokens:

$$(\mathbf{K}_\ell^{\text{cur}}, \mathbf{V}_\ell^{\text{cur}}) = \text{Prefill}\left(f_\theta, q, I\right). \tag{24}$$

We then form the augmented KV sequences

$$\tilde{\mathbf{K}}_\ell = \text{concatseq}\left(\mathbf{K}_\ell^{\text{ext}}, \mathbf{K}_\ell^{\text{cur}}\right), \tag{25}$$

$$\tilde{\mathbf{V}}_\ell = \text{concatseq}\left(\mathbf{V}_\ell^{\text{ext}}, \mathbf{V}_\ell^{\text{cur}}\right). \tag{26}$$

Self-attention at layer $\ell$ is computed as

$$\mathbf{A}_\ell = \text{softmax}\left(\frac{\mathbf{Q}_\ell \tilde{\mathbf{K}}_\ell^\top}{\sqrt{d_k}} + \mathcal{M}\right) \tilde{\mathbf{V}}_\ell, \tag{27}$$

where $\mathbf{Q}_\ell$ is the layer-$\ell$ query projection, $d_k$ is the key dimension, and $\mathcal{M}$ is the attention mask (external prefix fully visible; current tokens autoregressive). Relative or rotary position encodings are offset by $L_{\text{ext}}$ to maintain consistency after concatenation.

If $\mathcal{S}(q, I)$ changes across turns, we incrementally prefill only the missing concepts to extend the external cache without recomputing previously stored $\text{KV}^{(c)}$. The final output of the model can be written as

$$y = \text{Decode}\left(f_\theta; \{\text{KV}^{(c)}\}_{c \in \mathcal{S}(q,I)}, q, I\right), \tag{28}$$

where Decode denotes a standard single-pass transformer decoder conditioned on both the external KV cache and the current input tokens.

# E   CONSTRUCTION OF YO'LLAVA++ AND MC-LLAVA++

This section explains how we construct the Yo'LLaVA++ and MC-LLaVA++ datasets and gives the exact prompt templates and JSON formats used during generation. The key idea is to work at the level of concepts. Each concept is an entity about which we generate instance specific binary questions. Yo'LLaVA++ and MC-LLaVA++ are produced by the same concept level pipeline. The only differences are the underlying image collections and the choice of the identifier token that appears in the questions. All other steps, including patch selection, prompt construction, question parsing, and answer rebalancing, are identical.

### E.1 Unified concept setting

We consider a set of concepts $\mathcal{C}$. For each concept $c \in \mathcal{C}$, the generation process assumes the following inputs, some of which are optional.

- A concept identifier token $t_c$ that appears in every question. In the concept level setting this is the textual concept id, for example `concept_123`. In the per image setting we use a fixed anonymous token such as `<sks>`.

- A set of one or more RGB images $\mathcal{I}_c$ associated with $c$. In the concept level setting, $\mathcal{I}_c$ collects all images for that concept. In the per image setting, $\mathcal{I}_c$ usually contains a single image from an existing VQA dataset.

- An optional information record that contains a short concept description (field `concept`) and a list of fingerprint attributes (field `fingerprint_attributes`) that describe distinctive local traits.

- An optional hard region metadata file that stores characteristic bounding boxes in a canonical resolution for each concept. Each entry specifies the image path, a bounding box $\{x, y, w, h\}$ in a base resolution `base_wh`, and is used to crop concept only patches.

We apply one unified construction procedure to these inputs, and obtain a set of A/B questions $\mathcal{Q}_c$ for each concept $c$.

### E.2 Unified construction procedure

Given the inputs above, we generate questions independently for each concept.

For a concept $c$, we first select a small visual context set $V_c$. If hard region metadata is available, we crop all annotated regions, resize them to a fixed resolution, and obtain a pool of concept only patches. When fingerprint attributes and a CLIP compatible encoder are both available, we embed the attributes and patches, compute cosine similarities, and keep patches whose similarity to any attribute is below a threshold. These patches tend to capture harder visual details that are not trivially explained by a single attribute. We then subsample at most $N_{\text{img}}$ patches. If no hard region metadata is available, we randomly subsample at most $N_{\text{img}}$ full images from $\mathcal{I}_c$ and resize them. The selected patches or images form $V_c$ and are only used as visual input to the multimodal model. They are never mentioned explicitly in the generated text.

Next, we build textual guidance. We turn the fingerprint attributes into short trait anchors by stripping boilerplate prefixes and truncating the list to a fixed length. When hard region metadata is present, we also compute simple statistics for each patch in $V_c$, including dominant colour, a coarse texture tag, and a rough body location such as left, center, or right. We convert these statistics into hard region hints, which are short internal cues such as "right torso flank region shows cream, soft fur". Together, the trait anchors and hard region hints summarize concept specific evidence without exposing raw coordinates.

We then construct an instruction prompt for concept $c$. In the concept level setting, the prompt introduces the identifier token $\{$`concept_id`$\}$, lists the trait anchors and hard region hints, and states strict rules. Questions must be individual specific, must mention the identifier token, must avoid deictic wording and camera terms, must not describe backgrounds or generic anatomy, and must follow a binary A/B format with word limits and a fixed JSON schema. In the per image setting, we use a similar prompt but refer to the concept only through a fixed anonymous token such as `<sks>` and allow questions to depend on both the concept and visible objects in the scene.

Finally, we send the prompt together with $V_c$ as a single multimodal input to the language model and request a JSON object. We parse the returned JSON, discard entries that are structurally invalid, truncate overly long questions, and enforce that every question uses the chosen identifier token, either the concrete concept id or the anonymous token. We remove exact duplicates, keep at most a target number of questions per concept, and require a minimum number of valid items. We then rebalance the proportion of correct answers A and B by swapping option texts in a small number of questions when necessary. The remaining items form the question set $\mathcal{Q}_c$ for concept $c$.

### E.3 PROMPT TEMPLATES AND QA FORMAT

---

**Concept level instance specific QA prompt**

**Role.** You are a vision language analyst. You see several reference images of the same concept, denoted by the token {concept_id}. Your goal is to create concept specific binary questions about this concept.

**Input.** A small set of reference images of {concept_id}, a brief concept description if available, a bullet list of trait anchors {trait_anchors_bulleted}, and a bullet list of hard region hints {hard_region_hints_bulleted} derived from characteristic concept only patches.

**Hard constraints.** Do not use deictic terms such as "this image", "this picture", "here", "above", or "below". Do not use camera or cropping words such as "crop", "patch", "box", "frame", "zoom", or "close up". Do not describe backgrounds or scenes such as sky, grass, road, table, wall, floor, or furniture. Do not ask species or breed level questions. Do not ask trivial anatomy questions such as the number of legs or the presence of fur.

**Target style.** Each question must depend on the individual specific traits of {concept_id}. Use trait anchors and hard region hints as inspiration but do not copy them word for word. Focus on fine grained local cues, left right asymmetries, small discrete counts, and stable micro patterns on the concept. Use English only, in a concise telegraphic style.

**Question and option constraints.** Each question must contain the token {concept_id} and must have at most 18 words. Each option must contain at most 3 words. Provide two options per question, labeled "A" and "B". Both options must be plausible. Across the entire set, answers should be roughly balanced between A and B.

**Output schema and cardinality.** Return a single JSON object whose keys are string indices "0", "1", and so on. Each key maps to one item with the following fields:

```
{"question":  "string",
  "option":  {"A":  "string", "B":  "string"},
  "correct_answer":  "A" or "B"}
```

Generate {q_target} questions in total if possible, with at least {q_min} structurally valid items.

---

**Per image QA prompt with anonymous concept token**

**Role.** You are a vision language analyst. You see one image containing a concept that is referred to by the literal token <sks>. You generate binary questions that are answerable from this image alone.

**Input.** One RGB image. The concept in the image must always be called <sks>. No other name is allowed.

**Hard constraints.** Do not use deictic terms such as "this image", "this picture", "here", "above", or "below". Do not use camera or cropping words such as "crop", "patch", "box", "frame", "zoom", or "close up". Do not invent any name other than the literal token <sks>.

**Target style.** Questions may rely on both the concept and visible objects or layout in the scene. Use only information that is visible in the image and avoid external knowledge. Questions must be concise, visually grounded, and written in English.

**Question and option constraints.** Always use the token <sks> whenever you refer to the concept. Each question must have at most 18 words. Each option must have at most 3 words. Provide two options per question, labeled "A" and "B". Both options must be plausible. Across the set, answers should be approximately balanced between A and B.

**Output schema and cardinality.** Return a single JSON object whose keys are string indices "0", "1", and so on. Each key maps to one item with the structure:

```
{"question":  "string",
  "option":  {"A":  "string", "B":  "string"},
  "correct_answer":  "A" or "B"}
```

Generate exactly {n} questions for the given image.

---

**QA JSON format in Yo'LLaVA++ and MC-LLaVA++**

**Per question format.** Each question instance in both datasets follows the same JSON schema:

```
{"question":  "For concept_123, which ear has the larger dark
patch?",
  "option":  {"A":  "left ear", "B":  "right ear"},
  "correct_answer":  "B"}
```

**Dataset layout.** At storage time, we map each concept identifier to its question entries. In the concept level setting, the top level maps each concept id to a dictionary of indexed question entries. In the per image setting, the top level maps each concept id to a dictionary from image path to a list of indexed question entries. In all cases, the inner question objects follow the schema above.

## F   TEXT METADATA CONSTRUCTION WITH A VISION–LANGUAGE API

### F.1   OBJECTIVE

We construct, for each concept (object or person), a compact JSON record with exactly four fields: `concept`, `category`, `caption`, and `fingerprint_attributes`. The goal is to support downstream retrieval, grounding, and patch-level reasoning with minimal yet highly discriminative metadata.

### F.2   INPUT ORGANIZATION

Images are organized under a root directory where each subfolder name serves as a concept identifier. Every subfolder contains one or more photos related to the same concept. Standard formats are accepted (JPEG, PNG, WebP). The subfolder name provides a stable concept hint used to seed the textual fields when the model response omits them.

### F.3   IMAGE PREPROCESSING

Each image is opened and converted to RGB if necessary. To control bandwidth and ensure consistent visual quality, the long side is constrained to a maximum width of 2048 pixels, while preserving the aspect ratio. The image is then encoded as JPEG with a quality of 92. The encoded bytes are then base64-encoded and embedded into the chat request as data URLs. This keeps the entire interaction self-contained, eliminating the need for external hosting.

### F.4   API REQUEST COMPOSITION

For each concept, the request payload contains a single textual instruction (the full prompt is provided verbatim later in this subsection) followed by all images of the concept, each attached as an image block via a data URL. The request is issued to a multimodal chat completion endpoint with the following critical settings: JSON-only response enforcement, deterministic decoding, and conservative generation length. Concretely, the generation uses a low temperature (0.0), an explicit request for a JSON object as output, a token limit sufficient to cover the required fields, and a fixed seed. To improve robustness under transient network or service conditions, the client implements bounded exponential backoff with jitter for a small number of retries in response to rate limiting or server errors.

### F.5 RESPONSE PARSING

Since models sometimes wrap JSON in explanatory text or code fences, the response is sanitized before parsing. The parser removes code fences if present and scans for the outermost well-formed JSON object. Only that object is then parsed. If no valid object is found, the record for that concept defaults to a minimal placeholder, logging a parsing error for inspection.

### F.6 SCHEMA ENFORCEMENT AND NORMALIZATION

The post-processor guarantees that the final record contains exactly the four fields in the expected order. Missing values are imputed as follows: the concept name defaults to the subfolder name when absent; the category and caption default to `unknown`. The attribute list is normalized to a unique, order-preserving list; if empty, a single placeholder ``unknown'' is inserted to avoid downstream edge cases. To keep the representation lightweight, the attribute list is capped at a modest upper bound. The output map across all concepts is serialized as human-readable UTF-8 JSON with indentation, preserving non-ASCII characters.

### F.7 OPERATIONAL DETAILS

Command-line arguments control the image root, output path, model identifier, API base URL, and API key. The API key can be passed via argument or read from an environment variable. Informational logging summarizes progress per concept and reports any parsing or I/O anomalies. Practically, the pipeline tends to be I/O-bound on large folders, while the inference cost scales with image count and attribute richness; the enforced JSON and normalization keep the downstream footprint predictable.

### F.8 FULL PROMPT USED FOR METADATA CONSTRUCTION

The exact instruction sent to the model is reproduced below in verbatim form.

---

**Four-Key Metadata Prompt (Full)**

**Role.** You are a vision–language analyst. You will see multiple images of the **same concept** (same object/person). Produce a minimal JSON.

**Output schema (exact keys, exact order).**

```
"concept": "string", "category": "string", "caption": "string",
"fingerprint_attributes": ["...", "..."]
```

**Global style (English only).** Telegraphic style, commas and hyphenated compounds, lowercase everywhere except exact wordmarks/logos (keep casing), no periods, separate clauses with commas.

**Evidence policy (silent).** Use visual evidence only. Tally traits per image; include traits present in $\geq 60\%$; prefix `often:` for 40–60%; never include $< 40\%$. Resolve conflicts by majority with ties toward more discriminative cues (shape < color/pattern < material < background). Include background only if stable & discriminative. Allow negative but discriminative cues with `none`/`absent` (e.g., `tail: none`, `logo: absent`). Do not infer brand/material if uncertain. If identity/name is unknown, do not guess.

**Field rules.**

- **concept (3–7 tokens):** signature phrase aiding retrieval & disambiguation; include $\geq 1$ discriminative token (breed/shape/color/role); avoid verbatim reuse of any `[part]: [descriptor]`; examples: `shiba inu plush`, `curled-tail gray`, `cat ceramic mug`, `male adult with glasses`; if unclear write `unknown`.

- **category (1–3 tokens):** short normalized label, open-world; may use simple hierarchy with < (e.g., `animal<dog`, `device<keyboard>`); pick the most defensible label; avoid long phrases and overfitting to fine-grained names when uncertain.

---

- **caption (24–30 words):** concept-level majority summary in strict order—(1) silhouette/shape, (2) dominant colors/patterns, (3) two–three signature parts, (4) stable accessory/wordmark/-material. Do **not** include `often:`; do **not** copy any exact `[part]: [descriptor]` string; paraphrase at a higher level; optionally append `palette: X dominant; Y secondary`.

- **fingerprint_attributes (15–16 items):** each $\leq 6$ words, pattern `[part]: [descriptor]`; may prefix `often:` for 40–60% traits; cover $\leq 8$ distinct parts; $\leq 2$ items per part unless clearly distinct & discriminative; order by utility—positions 1–6 hard-localizable parts (best for patching/grounding), 7–12 global appearance/pose/material, 13–16 auxiliary/negative/background (logos/wordmarks/stable background). Prefer attributes that directly answer likely QA.

**Descriptors & vocab (preferred, open-world allowed).**

- **Colors:** black, white, gray, silver, gold, red, orange, yellow, green, blue, purple, pink, brown, beige, navy, teal; concise free forms allowed (e.g., `blue-green`, `warm gray`).

- **Patterns:** solid, stripes, polka dots, plaid, floral, check, camo, gradient, logo, wordmark; concise free forms allowed.

- **Shapes:** round, square, rectangular, oval, triangular, tapered, curved, flat, ribbed, ridged, beveled, domed; concise free forms allowed.

- **Materials:** cotton, denim, leather, metal, plastic, wood, ceramic, glass, fabric, rubber, fleece, knit, plush; concise free forms allowed.

- **Accessories:** glasses, hat, cap, bow, scarf, helmet, tie, lanyard, headphones.

- **Species/breed tokens:** when visually warranted are allowed but not required (dog, cat, shiba inu, husky, corgi, ragdoll, british shorthair).

**Parts vocabulary (singular, reusable).**

- **Core parts:** hair, ear, eye, face, beard, whiskers, muzzle, tail, collar, chest, back, belly, paw, hand, beak, horn, fin, wing, handle, lid, rim, body, keycap, roof, spire, window, door, levels, inscription, strap, pocket, sleeve, cuff, collarband, button, zipper, logo, wordmark, glasses, hat, bow, scarf, helmet, tie, lanyard, headphones, breed, species, pose, gesture, background, base.

- **Extension rule:** If a salient part is missing, introduce a concise new singular noun and reuse it consistently.

**Coverage guidance (use when applicable).**

- **Animals/toys:** species/breed, dominant coat colors, white/black presence, eye color, whiskers, tail presence/curvature, collar presence, material, pose, cartoon/humanlike if applicable.

- **Cups/mugs:** body shape, handle presence, material, rim, lid presence, motif/face cues, dominant color, exact wordmark text when present.

- **Devices:** body shape, body color, keycap color/pattern, logo/wordmark when present.

- **Buildings/landmarks:** primary color/material, roof presence, door/window, spire count, levels count, style cue, inscription presence, base/platform, background.

- **Persons:** glasses, hair length/texture/color, beard/bald, adult/child, top type & color/pattern, accessories, gestures, visible wordmark/logo/number on clothing, background.

**Attribute formatting examples (illustrative).**

- **Animal/toy:** `species: dog, breed: shiba inu, tail: curled, collar: present, whiskers: present, body: orange solid, material: plush`

- **Cup:** `body: round, handle: present, material: ceramic, rim: domed, wordmark: Neural Information Processing Systems`

- **Person:** `glasses: present, hair: black short, beard: absent, top: black solid, logo: Microsoft, gesture: peace-sign, background: lake`

- **Building:** `spire: count-2, levels: count-7, roof: present, inscription: present, base: rocks, body: gothic`

**Quality & count checks (silent, mandatory).**

- **Output JSON only:** no prose, no code fences, no extra keys, no trailing commas.
- **Exactly four keys:** in the required order.
- **Caption length:** 24–30 words; ordering & paraphrasing rules respected; optional palette ranking allowed.
- **Fingerprint size:** `fingerprint_attributes` has 15–16 items; each $\leq 6$ words; $\leq 8$ parts; $\leq 2$ per part unless clearly distinct & discriminative; positions 1–6 are hard-localizable cues.
- **Thresholds:** majority thresholds & tie-breaking by discriminativeness respected; exact text preserved for wordmarks/logos; lowercase otherwise.
- **Unknown concept:** If concept name is uncertain, set `"concept": "unknown"` but still complete caption and attributes from majority-evident traits.

**Final output requirement.** Return the final JSON **only**, conforming to all rules above.

**Discussion** The instruction biases the model toward compact, discriminative, and patch-friendly attributes. The majority-threshold policy, negative evidence allowance, and strict formatting constraints jointly reduce hallucination and enforce consistent field semantics across concepts. The preprocessing and JSON normalization further ensure that downstream retrieval and grounding operate over a predictable schema with bounded size and stable key ordering.

## G EXPERIMENTAL SETUP AND HYPERPARAMETERS

### G.1 BASELINES

For Yo'LLaVA, we adopt the LLaVA-OneVision-Qwen2-7b VLM backbone and train 16 soft tokens per subject using 5–10 positive images and approximately 150 hard negatives retrieved per subject. Each subject is taught for up to 15 epochs, and the best checkpoint is selected based on recognition accuracy on the training split. For RAP-MLLM, we utilize the RAP-LLaVA-13b model provided by the original authors. For each subject, we randomly select one training image as the avatar and sample several images to prompt GPT-5 to generate a description.

### G.2 EVIDENCE CONSTRUCTION

Unless stated otherwise, each concept uses $k = 5$ evidence images to synthesize a textual profile (canonical name, category, $\sim$25-word caption, and fingerprint attributes) via the GPT-5 API with deterministic decoding (temperature $= 0$) (OpenAI, 2025). Hard patches are mined per concept using Algorithm 1 with grid size $g = 12$, fusion exponent $\gamma = 1$, background suppression enabled, and top-$k = 4$ patches retained. Text fields are tokenized once to build the concept-specific external KV cache; hard patches populate a compact visual index.

### G.3 IMPLEMENTATION DETAILS

All methods share the same backbone (LLaVA-OV) and tokenizer. Generation uses greedy decoding unless otherwise noted. KV caches store FP16 tensors and persist per concept. Image resolution, preprocessing, and vision-tower normalization follow the backbone defaults for fairness.

### G.4 EVALUATION PROTOCOL

For each concept: (i) sample $k$ evidence images to build a text profile and mine candidate hard patches; (ii) construct a single text KV cache; and (iii) index all candidate patches. At inference time, embed the query, compute its similarity to all attribute strings, and select the top-$k$ attributes for that turn; the selected attributes and their corresponding top-$k$ visual patches form the evidence fed to the model. In multi-turn sessions on the same concept, cached `past_key_values` are reused to avoid re-prefill.

## H EXTENDED RELATED WORK

### H.1 MULTIMODAL LLMS

Multimodal large language models (LMMs) have advanced general perception and open-ended reasoning by pairing stronger visual encoders with instruction tuning and extensive training on multi-image and video data. In parallel, multi-step visual reasoning with token scaling and verification offers a complementary path toward more reliable inference (Bai et al., 2025b). Representative systems include LLaVA-OneVision, which unifies single- and multi-image and video transfer (Li et al., 2024a), InternVL-2.5, which scales to extensive open benchmarks (Chen et al., 2024), and Qwen2.5-VL, which emphasizes precise localization and long-context parsing (Bai et al., 2025a). Recent foundations further consolidate design choices for modern vision–language models (VLMs): MM1 conducts extensive architecture and data ablations (McKinzie et al., 2024); MM1.5 carries these insights into finetuning (Zhang et al., 2025); Idefics2 offers a practical recipe for grounding and multi-image dialogue (Laurençon et al., 2024); Molmo provides open weights and open data (Deitke et al., 2024); and LLaVA-NeXT-Interleave supports multi-image, video, and 3D via interleaved formats (Li et al., 2024b), while advances in multi-granularity video representation further strengthen the video pathway for MLLMs (Shi et al., 2025). Despite these developments, current models in practice still struggle to reliably represent persistent, user-specific concepts, which motivates the need for explicit personalization mechanisms.

### H.2 PERSONALIZATION VIA PARAMETER ADAPTATION

One research line personalizes models by updating parameters or attaching lightweight modules for each concept. (Nguyen et al., 2024) learns compact subject tokens from a few images. (Nguyen et al., 2025) extends from recognition to generation under few-shot constraints. (An et al., 2025a) unify learned concept tokens to facilitate understanding and generation within a single interface. (An et al., 2025b) organizes synthetic expansions from a small number of seeds to encompass attributes and contexts. (An et al., 2024) addresses realistic multi-concept composition using instruction tuning and personalized prompts. Beyond these personalized methods, the broader parameter-efficient toolbox reduces per-concept training costs, including LoRA (Hu et al., 2022), prefix/prompt tuning (Li & Liang, 2021), P-Tuning v2 (Liu et al., 2021), and Visual Prompt Tuning (Jia et al., 2022), as well as Visual Prompt-based instruction for MLLMs (Lin et al., 2024). While effective in accuracy, these approaches still require maintaining adapted artifacts for each user or concept, which complicates versioning and deployment.

## I LIMITATIONS OF PRECOMPUTED EXTERNAL KV AND FUTURE DIRECTIONS

Our design treats precomputed external KV as a central personalization primitive. This brings clear benefits in efficiency and stability, but also introduces limitations that are important to acknowledge, especially for complex personalized queries that require reasoning beyond the stored evidence.

First, precomputed KV is fundamentally a *frozen* representation of the available concept evidence. It is well suited for questions that can be answered by selectively attending to the stored attributes and hard patches, but less suited for queries that require multi-step reasoning, composition with new facts, or conflict resolution across concepts. In such cases, the model still relies on its backbone knowledge and on-the-fly inference, while the external KV only provides a fixed set of anchors. This means that, for highly abstract or long-horizon personalized tasks, our method may not fully capture the adaptation capacity of approaches that update parameters or continuously refine their memories.

Second, our current pipeline assumes that the concept evidence itself is reasonably complete and clean. The external KV caches are built from a finite set of images and attributes; if these are biased, partially incorrect, or miss entire modes of appearance, the resulting personalization will inherit these gaps. Since the KV states are precomputed and reused across sessions, such biases are not corrected automatically by interaction. This is a general limitation of static memory based personalization and is not unique to our approach, but becomes more visible when the KV cache is treated as the main personalization channel.

Third, the retrieval stage is deliberately compact. We constrain the number of retrieved attributes and hard patches by top-$k$ thresholds in order to keep decoding efficient. While Section **??** shows that performance is stable in a reasonable range of $k$ values, this design also means that we do not expose all potentially relevant evidence to the model for very complex queries. For example, queries that span multiple concepts, or that require reconciling rare visual configurations, might benefit from a more adaptive retrieval budget than the fixed choices used in this work.

These limitations point to several promising directions for improving adaptability:

- **Adaptive and query-aware KV refinement.** Instead of using a single precomputed KV per concept, one could consider a two-stage scheme where coarse external KV is retrieved first, then refined or expanded on-the-fly based on the current query and intermediate activations of the backbone.

- **Hybrid designs that combine external KV with lightweight updates.** Our method keeps the backbone strictly frozen. A natural extension is to allow small, concept-specific adapters or low-rank updates that are conditioned on the retrieved KV, so that the model can better integrate stored evidence with new reasoning patterns without full fine-tuning.

- **Richer uncertainty and conflict handling.** For queries that go beyond the support of the stored evidence, the model should be able to detect when the retrieved KV is insufficient or contradictory, and either fall back to the base model or explicitly signal uncertainty. This could be implemented by monitoring retrieval scores or by learning simple confidence heads on top of the decoder states.

- **Incremental updates to the external memory.** Finally, one could explore mechanisms for updating or augmenting the external KV cache over time as new images and interactions arrive, while keeping the overall memory compact. This would allow the system to gradually improve coverage of each concept without retraining the backbone.

We leave these extensions to future work. In the present paper, our focus is on showing that a purely precomputed, retrieval-indexed external KV can already provide strong and robust gains on fine-grained personalization benchmarks, even under these limitations.

## J  DISCUSSION ON MULTI-CONCEPT PERSONALIZATION

The main experiments in this work focus on single concept personalization, where each dialogue turn is conditioned on one resolved concept $c \in \mathcal{C}$ and its cached evidence $\big(T^{(c)}, P^{(c)}\big)$.

A natural question is how the same design extends to queries that involve multiple concepts, such as comparisons, joint reasoning, or conversations where the active identity changes over time. This section outlines how Jarvis can operate in a multi concept regime without changing the backbone or the evidence construction pipeline.

**Retrieval over multiple concepts.**   Assume the system maintains a global repository

$$\mathcal{R} = \big\{ \big(T^{(c)}, P^{(c)}, \mathrm{KV}^{(c)}\big) \mid c \in \mathcal{C} \big\}, \tag{29}$$

where $\mathrm{KV}^{(c)}$ denotes the precomputed external KV cache for concept $c$ obtained from its text prefix and mined hard patches. Given a user query $q_t$ and an optional image $I_t$ at turn $t$, Jarvis can rank all concepts by a joint retrieval score $s_{\mathrm{retr}}(q_t, I_t, c)$ that combines text similarity and hard patch similarity, exactly as described for the single concept case. Instead of resolving to a single concept, we select the top $K_{\mathrm{concept}}$ concepts

$$\mathcal{R}_t = \mathrm{TopK}\big\{ s_{\mathrm{retr}}(q_t, I_t, c) \,\big|\, c \in \mathcal{C} \big\}, \tag{30}$$

with a small $K_{\mathrm{concept}}$ (for example two or three) to control cache length.

**Cache composition across concepts.**   For the selected set $\mathcal{R}_t$, the external KV prefix for turn $t$ is formed by concatenating the corresponding caches along the sequence dimension

$$\mathrm{KV}_t^{\mathrm{ext}} = \bigoplus_{c \in \mathcal{R}_t} \mathrm{KV}^{(c)}, \tag{31}$$

followed by the current turn cache $\text{KV}_t^{\text{curr}}$ that encodes the user query and image. This is a direct generalization of the single concept setting, where $K_{\text{concept}} = 1$. The concatenation order can be made deterministic (for example by descending $s_{\text{retr}}$) to preserve reproducibility. Since each $\text{KV}^{(c)}$ is already compact and concept specific, the total prefix length grows roughly linearly with $K_{\text{concept}}$ and remains much shorter than rebuilding long multi concept prompts at every turn.

**Reasoning over relations between concepts.** Multi concept queries often ask for relations, such as comparisons of appearance or joint attributes. In Jarvis, such reasoning is still delegated to the frozen backbone: the external prefix supplies $K_{\text{concept}}$ separate evidence blocks, and the decoder attends over them when generating the answer. In practice this suggests two design principles. First, retrieval should be precise enough that irrelevant concepts rarely enter $\mathcal{R}_t$, otherwise cross concept interference may arise when several caches share similar attributes. Second, the answer template should encourage explicit comparison, for example by asking the model to describe each concept in turn and then summarize similarities and differences. Both principles can be implemented at the prompt level without new parameters.

**Open challenges and opportunities.** While the above extension already enables multi concept personalization with the existing infrastructure, it raises several research directions. One is cache routing: instead of a flat concatenation, future work may assign different concepts to different attention heads or layers, which would allow the model to specialize attention patterns for comparison versus recognition. Another direction is adaptive concept selection, where $K_{\text{concept}}$ is chosen based on a confidence estimate of $s_{\text{retr}}$, so that simple single concept queries do not pay the cost of extra caches. Finally, when the number of personalized concepts grows large, it becomes important to study memory management policies for eviction, compression, and privacy preserving storage. These questions go beyond the single concept setting studied in the main text, but they build directly on the same principle of materializing concept evidence as reusable KV and suggest a path toward scalable multi concept personalized assistants.

## K    ADDITIONAL QUALITATIVE RESULTS

In this section, we provide additional qualitative results in Table 11, which presents example responses from the Jarvis model, together with dataset examples in Tables 5–10, which showcase instances from our constructed Yo'LLaVA++ and MC-LLaVA++ datasets.

<cat-cup>: 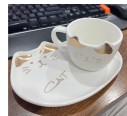

▷ *Yo'LLaVA++ (Text-only)*

| | | |
|---|---|---|
| Question: | Q1: Does <cat-cup> have a metallic brown sculpted ear? | Q2: Does <cat-cup> feature the word 'CAT'? |
| Question: | Q3: Is <cat-cup>'s nose small and black? | Q4: Does <cat-cup> have a matching cat motif saucer? |
| Question: | Q5: Is <cat-cup>'s handle absent? | Q6: Does <cat-cup> have an ear-shaped cutout on the rim? |
| Question: | Q7: Is <cat-cup> made of ceramic material? | Q8: Does <cat-cup> have a round body shape? |

Table 5: Examples from our Yo'LLaVA++ dataset, constructed from the concept <cat-cup>.

<mam>: 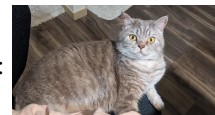

▷ *Yo'LLaVA++ (Text-only)*

| | | |
|---|---|---|
| Question: | Q1: Does <mam> have a darker ring around the amber iris? | Q2: Are <mam>'s long white whiskers denser on the left side? |
| Question: | Q3: Does <mam>'s silver gray tabby coat include brownish patches? | Q4: Is <mam>'s short broad muzzle marked with vertical stripes? |
| Question: | Q5: Are <mam>'s pointed ears tipped with darker fur? | Q6: Does <mam>'s medium thick tail have a uniform color? |
| Question: | Q7: Is the fur on <mam>'s left flank mottled gray? | Q8: Does <mam>'s broad round face have symmetrical markings? |

Table 6: Examples from our Yo'LLaVA++ dataset, constructed from the concept <mam>.

<lamb>: 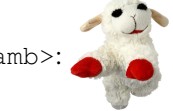

▷ *Yo'LLaVA++ (Text-only)*

| | | |
|---|---|---|
| Question: | Q1: Does <lamb> have beige floppy ears? | Q2: Is <lamb>'s stitched mouth black? |
| Question: | Q3: Are <lamb>'s hooves red solid? | Q4: Does <lamb> have a small round tail? |
| Question: | Q5: Is <lamb>'s body covered in white fleece? | Q6: Are <lamb>'s eyes black and oval? |
| Question: | Q7: Is <lamb>'s texture smooth instead of fluffy | Q8: Is <lamb>'s material plush? |

Table 7: Examples from our Yo'LLaVA++ dataset, constructed from the concept <lamb>.

<Eru>: 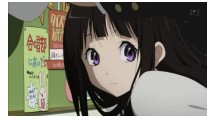

▷ *MC-LLaVA++ (Text-only)*

| Question: | Q1: Does <Eru>'s sailor collar have a visible blue bow? | Q2: Does <Eru>'s long hair appear straight? |
|---|---|---|
| Question: | Q3: Is <Eru>'s hair a light blonde shade? | Q4: Does <Eru> usually wear a white long-sleeve top? |
| Question: | Q5: Is <Eru>'s collar style round-necked? | Q6: Does <Eru>'s sailor collar have a visible blue bow? |
| Question: | Q7: Is <Eru> wearing any scarf around the neck? | Q8: Does <Eru> usually have a gesture of pointing? |

Table 8: Examples from our MC-LLaVA++ dataset, constructed from the concept <Eru>.

<Harry>: 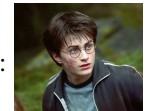

▷ *MC-LLaVA++ (Text-only)*

| Question: | Q1: Does <Harry> wear a collared shirt under his sweater? | Q2: Does <Harry> have a crest on his robe? |
|---|---|---|
| Question: | Q3: Are <Harry>'s glasses round or square? | Q4: Is <Harry>'s build slim or broad? |
| Question: | Q5: Is <Harry>'s hair medium or short length? | Q6: Is <Harry>'s tie striped or solid? |
| Question: | Q7: Does <Harry>'s tie have more red or more yellow? | Q8: Does <Harry> usually wear a collared shirt under his sweater? |

Table 9: Examples from our MC-LLaVA++ dataset, constructed from the concept <Harry>.

<HouTeng>: 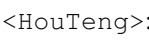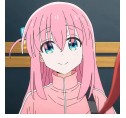

▷ *MC-LLaVA++ (Text-only)*

| Question: | Q1: Does <HouTeng> wear a blue or yellow square hairclip on the left side? | Q2: Is <HouTeng>'s guitar body black or red? |
|---|---|---|
| Question: | Q3: Are <HouTeng>'s eyes large blue or small green? | Q4: Is <HouTeng>'s hair long pink or short brown? |
| Question: | Q5: Does <HouTeng>'s guitar have a solid or transparent pickguard? | Q6: Does <HouTeng> have one or two visible hairclips? |
| Question: | Q7: Is <HouTeng> depicted in 2D animation or realistic style? | Q8: Does <HouTeng> have blush cheeks or none? |

Table 10: Examples from our MC-LLaVA++ dataset, constructed from the concept <HouTeng>.

Table 11: More examples of qualitative results.

| Concept | Conversation |
|---|---|
| 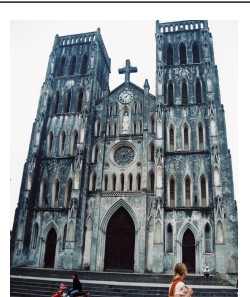
\<cat-cup\> | Question: What kind of ear does \<cat-cup\> have?
**Jarvis:** Metallic brown.

Question: What word is written on \<cat-cup\>'s matching saucer?
**Jarvis:** CAT.

Question: How would you describe \<cat-cup\>'s overall shape?
**Jarvis:** Cat Cup has a rounded mug silhouette. |
| \<nha-tho-hanoi\> | Question: How many floors does \<nha-tho-hanoi\> have?
**Jarvis:** Nha Tho Hanoi has four floors.

Question: How would you characterize the shape of \<nha-tho-hanoi\>'s roof?
**Jarvis:** Nha Tho Hanoi's roof has a flat top.

Question: What's special about \<nha-tho-hanoi\>'s architectural design?
**Jarvis:** Nha Tho Hanoi has a tall twin-tower silhouette, weathered gray stone facade, pointed arched windows and a large circular rose window, with a central cross and clock above the main entrance. |
| 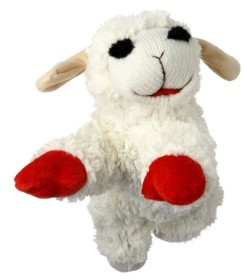
\<lamb\> | Question: Describe \<lamb\>'s ear.
**Jarvis:** Lamb's ear is beige and floppy.

Question: What's special about \<lamb\>'s appearance?
**Jarvis:** Lamb has a white fleece texture, beige ears, and bright red hooves.

Question: Does \<lamb\>have any accessory attached?
**Jarvis:** No, Lamb does not have any accessory attached. |