# OpenReview forum: "Jarvis: Towards Personalized AI Assistant via Personal KV-Cache Retrieval"
_ICLR.cc/2026/Conference — Submitted to ICLR 2026_

### Official Review · Reviewer_tLpQ · 2025-10-28

**Soundness:** 3
**Presentation:** 2
**Contribution:** 2
**Rating:** 4
**Confidence:** 4

**Summary:**

The paper introduces a framework named Jarvis for efficient VLM personalization using KV-Cache retrieval. It stores user-specific information in KV-Caches of both textual and hard visual tokens, which allows for accurate and efficient responses to personalized queries without the need for extensive fine-tuning. The authors also present a benchmark dataset for evaluating personalized VLMs on fine-grained visual question answering tasks. Experimental results show that Jarvis achieves good performance on both visual and text-only question answering tasks.

**Strengths:**

1. The idea of storing essential personalized information in a reusable KV cache is conceptually reasonable and can effectively reduce computational overhead during inference.
2. The framework is lightweight and training-free, which makes it practical and easy to deploy for large VLMs.
3. The experimental results show the effectiveness of the proposed framework on personalized VQA tasks.

**Weaknesses:**

1. Several implementation details remain insufficiently explained. For example, it is unclear how textual similarity is computed for retrieval, how the proposed Yo'LLaVA++ and MC-LLaVA++ datasets are constructed, and what the question formats in these datasets look like. The authors should provide more detailed descriptions in the main text or appendices to improve clarity and reproducibility.
2. The concepts of KV-cache and retrieval-based personalization have already been explored in prior works for both language and vision-language models. The paper should better clarify the technical differences between the proposed method and prior works, and ideally provide a fair empirical comparison to demonstrate its advantage.
3. Additional baselines (e.g., RePIC, PLVM, or other recent personalization methods) should be included to provide a more comprehensive evaluation.
4. The experiments are confined to question-answering tasks. It remains unclear how the proposed framework would perform on other multimodal tasks such as image captioning, recognition, or cross-concept interactions. The work also focuses only on single-concept personalization without discussing scalability to multi-concept personalization.
5. Important hyperparameters such as the number of retrieved patches (top-k) or the system's robustness to retrieval errors are not thoroughly analyzed.
6. The precomputed KV caches may fail to generalize to complex personalized queries that require reasoning beyond stored evidence. The authors should discuss this limitation and potential directions to improve adaptability.

**Questions:**

1. The evidence retrieval process appears to rely primarily on textual similarity. How does the system handle cases where the target concept is not explicitly mentioned in the query text?
2. Regarding the RAP-LLaVA baseline, is the model used in the experiments the officially released version, or was it retrained using the LLaVA-OneVision backbone for fair comparison? Clarifying this is important for assessing experimental consistency.
3. In Figure 4, why is MC-LLaVA not included in the latency and throughput comparison?

---

### Official Review · Reviewer_JB1P · 2025-10-30

**Soundness:** 3
**Presentation:** 3
**Contribution:** 3
**Rating:** 4
**Confidence:** 4

**Summary:**

The paper proposes Jarvis, a training-free framework for personalizing Vision-Language Models. The core contribution is a retrieval-centric approach that externalizes user-specific concepts into a pre-computed Key-Value (KV) cache. This cache, composed of textual metadata and discriminative visual patches, is injected at inference time to ground the model's output.
By doing so, the method circumvents the need for parameter updates or the reconstruction of lengthy prompts, which the authors claim yields significant improvements in latency and throughput. The empirical evaluation, conducted on several benchmarks including newly proposed fine-grained variants, reports state-of-the-art performance, particularly on tasks requiring detailed, attribute-level recall.

**Strengths:**

- The paper introduces Jarvis, a novel and practical training-free framework for personalization. The core idea of externalizing user-specific concepts into a reusable KV-Cache is elegant. It directly addresses the high latency and context-length limitations of prompt concatenation methods without the overhead of maintaining per-user model parameters.

-  The paper provides strong evidence of practical benefits, showcasing an order-of-magnitude reduction in latency and a corresponding increase in throughput (QPS). This efficiency makes the system viable for real-time, scalable deployment, a critical advantage for user-facing AI assistants.

**Weaknesses:**

- The experiments primarily focus on single-concept personalization within a given session. A key challenge for a real-world AI assistant is handling queries that involve interactions between multiple personalized concepts (e.g., "my dog," "my daughter," "my car"). It is unclear how the proposed KV-Cache retrieval and concatenation would scale in complexity and performance when a query ambiguously references several distinct entities.

- The paper presents evidence construction as a one-time, offline process. However, user-specific concepts are often dynamic; users provide new images or information over time. The paper does not discuss a mechanism for efficiently updating or augmenting the KV-Cache for an existing concept without re-running the entire heavy offline pipeline. This is a practical limitation for long-term personalization.

**Questions:**

Plz answer my questions in the weakness section

**Details Of Ethics Concerns:**

The paper proposes Jarvis, a training-free framework for personalizing Vision-Language Models. The core contribution is a retrieval-centric approach that externalizes user-specific concepts into a pre-computed Key-Value (KV) cache. This cache, composed of textual metadata and discriminative visual patches, is injected at inference time to ground the model's output.
By doing so, the method circumvents the need for parameter updates or the reconstruction of lengthy prompts, which the authors claim yields significant improvements in latency and throughput. The empirical evaluation, conducted on several benchmarks including newly proposed fine-grained variants, reports state-of-the-art performance, particularly on tasks requiring detailed, attribute-level recall.

---

### Official Review · Reviewer_67Ra · 2025-10-30

**Soundness:** 3
**Presentation:** 3
**Contribution:** 3
**Rating:** 6
**Confidence:** 4

**Summary:**

this paper propose Jarvis, which is a personalized vlm model. it stores user-specific information as external key-value (KV) caches, which are precomputed and reused across turns without modifying the base model parameters. ​ This reduces latency and token usage while maintaining accuracy and responsiveness. ​

the main idea is that: user-specific information is summarized into compact text metadata and visual patches. ​ and the text metadata includes a canonical name, category, caption, and fingerprint attributes, while visual patches are mined from user images using techniques like GroundingDINO and SAM to extract discriminative regions.

At inference, Jarvis retrieves relevant evidence from the KV cache based on the user query and attaches it as external KV to the model. ​ This ensures responses are grounded in user-specific details without relying on lengthy prompts. ​

The authors introduce a benchmark that emphasizes accurate question answering based on fine-grained user-specific details, such as localized visual cues and abstract features

**Strengths:**

I like the idea how they reduced latency: by precomputing and reusing external KV caches, Jarvis significantly reduces latency and improves throughput, making it suitable for real-time applications

The framework achieves state-of-the-art results in both text-only and visual question answering tasks, particularly excelling in fine-grained, user-specific scenarios. ​

**Weaknesses:**

Personalized VLMs have been extensively studied in recent years; therefore, the overall scope of this paper feels somewhat limited. I encourage the authors to propose something more novel or distinctive.

the approach and design is a bit complicated, which not sure about scalability.

**Questions:**

na

---

### Official Review · Reviewer_DAdL · 2025-10-30

**Soundness:** 2
**Presentation:** 1
**Contribution:** 2
**Rating:** 2
**Confidence:** 4

**Summary:**

This paper proposes Jarvis, a method for addressing the problem of personalized LVLM. Jarvis is a retrieval and caching method for LVLM personalization, which retrieves the relevant information from the concepts corresponding to the user query and uses a caching method to insert it into the Query and Key cache of the LVLM model. Compared with training methods like YoLLaVA and MC-LLaVA, Jarvis can enable the personalization of LVLM without test-time fine-tuning.

**Strengths:**

1. The proposed framework is lightweight for LVLM personalization. It is a training-free method that addresses the problem of LVLM personalization without spending a brutal concept embedding training in YoLLaVA and MC-LLaVA.
2. The experiments indicate the superior performance of Jarvis compared to the personalization baselines.

**Weaknesses:**

There are several weaknesses that should be addressed in this paper:
1. Clarity: This paper does not have a clear presentation, especially in the Method presetation. In Algorithm 1, there are several terms that confuse the reader, including: box $B_m(u,v)$ (Line 287), OpenCLIP Relevance $R_m^{+}$ (Line 282), why we have the background map be the set $\{R_{m,b}^{-}\}_{b\in\mathcal{B}}$ (Line 283).
2. The motivation behind the algorithm is not clearly explained: For example, Algorithm 1 contains a term called the difficulty map $C_m$, but it is not clearly explained in the main paper why taking the Diffusion Inversion can result in a difficulty map. Other terms such as background negative $\mathcal{B}$ are not clearly explained in Algorithm 1. Also, why is the motivation behind the update in Lines 6, 10, and 11 in Algorithm 1?
3. Restricted only to single concept personalization: Jarvis only addresses the problem of Single concept personalization. However, it is running on LLaVA-OV, which can accept multiple input images to the input, thus having the capability of multiple-concept personalization in the architecture of LLaVA-OV. In addition, conducting only single-concept personalization is not convincing, since most of the dataset for single-concept personalization contains only a single object of that concept, thus leading to the scenario that the LVLM can respond to the query input, while not being aware of what the user is referring to.
4. In scenario 2 of Figure 1, the paper claims that Jarvis can clearly highlight the difference. However, from the answer, we do not know what the usual clothes <viruss> are. The difference between RAP-LLaVA and Jarvis only comes from "Different from usual casual
attire"; however, it does not indicate that RAP-LLaVA cannot follow the instruction. Maybe with a different seed, the response of RAP-LLaVA includes the term "Different from usual casual attire".

**Questions:**

Based on the weaknesses, there are several follow-up questions to address my concerns:
1. What is the definition of some terms: background negative $\mathcal{B}$, background map $R_{m, b}^{-}$ in Algorithm 1? In this case, the paper should give a clearer definition of these terms, because it is hard to follow and understand the patch mining presented in Algorithm 1.
2. Why does the difficulty map relate to the Stable Diffusion Inversion? What is the motivation in the equation at Lines 6, 10 and 11 in Algorithm 1?
3. In Appendix D, what is the linearization of $T^{(c)}$ into short prefix $\tau^{(C)}$ (Line 657), and how do we retrieve the small concept $S(q,I)$ in Line 668?
4. Do we add patch $P^{(c)}$ for each concept to the LLaVA-OV in test time, and how do we add that? I cannot find that in the paper. Why do we need to select the top-k element in Algorithm 1? Do we conduct an ablation study for k?

---

### Author Response · Authors · 2025-11-20
**Response to Reviewer tLpQ (W6)**

# Response to Reviewer tLpQ (W6)

---

## W6. Limitations of precomputed KV caches and adaptability

> *The precomputed KV caches may fail to generalize to complex personalized queries that require reasoning beyond stored evidence. The authors should discuss this limitation and potential directions to improve adaptability.*

**Response.**

We appreciate this insightful comment and fully agree that precomputed KV caches are not a universal solution for all forms of personalization. In the revised manuscript we now explicitly discuss this limitation and outline several directions to improve adaptability in a dedicated Appendix I. Here we summarize the main points.

---

### Where precomputed external KV is strong and where it is limited

Our design is tailored for fine-grained identity understanding, where the main need is to reliably ground the model on a concept’s stable visual and textual fingerprints. In this regime, a compact, precomputed external KV works very well: it provides high-precision evidence, is easy to retrieve, and can be reused across sessions without any backbone updates.

However, we acknowledge that this design has clear limitations:

- The external KV is **frozen** once built. It is well matched to questions that can be answered by attending to the stored evidence, but less suited to long-horizon or highly abstract personalized queries that require multi-step reasoning, combining new facts, or resolving conflicts between concepts.
- The quality of personalization is **bounded by the evidence**. If the underlying images and attributes for a concept are biased, incomplete, or noisy, the precomputed KV cache will inherit these issues and cannot correct them on its own.
- Our current retrieval budget (top-$k$ attributes and top-$k$ patches) is **fixed and conservative** by design, in order to keep decoding efficient. This means we do not always expose all potentially relevant evidence for very complex queries.

These limitations do not affect the core claims of the paper, which focus on fine-grained identity benchmarks, but they are important when thinking about broader, open-ended personalization.

---

### Potential directions to improve adaptability

Motivated by the reviewer’s concern, we have added a discussion of several high-level extensions that could increase adaptability beyond what is explored in this work:

- **Adaptive and query-aware KV refinement.**
  Instead of using a single static KV block per concept, one could first retrieve a coarse KV, then refine or expand it based on the current query (for example, by running an extra lightweight prefill conditioned on both the retrieved KV and the query).

- **Hybrid designs combining external KV with small updates.**
  Our method keeps the backbone strictly frozen. A natural extension is to add tiny concept-specific adapters or low-rank updates that are modulated by the retrieved KV, so that the model can gradually adjust its reasoning patterns while still leveraging the compact external memory.

- **Better handling of uncertainty and conflict.**
  When retrieved evidence is insufficient or internally inconsistent, the model should be able to detect this and either fall back to the base VLM or explicitly express uncertainty, rather than over-committing to the stored KV. This could be done by monitoring retrieval scores or adding simple confidence heads.

- **Incremental updates to the external memory.**
  Over time, as new images or interactions become available, one could update or augment the KV cache for each concept, allowing the memory to improve coverage without full retraining of the backbone.

We view these directions as promising next steps and believe they complement, rather than contradict, our current contribution. In this paper we focus on showing that even a purely precomputed, retrieval-indexed external KV already brings strong and robust gains on fine-grained personalization benchmarks; extending this design to more adaptive and interactive settings is an important avenue for future work.

---

### Author Response · Authors · 2025-11-20
**Response to Reviewer tLpQ (W5)**

# Response to Reviewer tLpQ (W5)

---

## W5. Ablations on top-k retrieval hyperparameters

> *Important hyperparameters such as the number of retrieved patches (top-k) or the system's robustness to retrieval errors are not thoroughly analyzed.*

**Response.**

We appreciate the reviewer’s comment. Our pipeline contains three top-$k$ choices: the number of retrieved attributes $k_{\text{attr}}$, the number of retrieved visual patches at test time $k_{\text{vis}}$, and the number of mined hard patches per image when building the index $k_{\text{patch}}$. All three serve as controlled compression knobs: they limit how many textual or visual items are exposed to the retriever and decoder, trading off coverage, robustness, and efficiency.

In the revised manuscript we now provide explicit ablations over these hyperparameters in Figure 5(a) and Figure 5(b), and we summarize the observations below.

---

### Ablation on $k_{\text{attr}}$ and $k_{\text{vis}}$ (Figure 5(a))

Figure 5(a) studies the interaction between the number of retrieved attributes $k_{\text{attr}}$ and the number of retrieved visual patches $k_{\text{vis}}$ at test time, evaluated on text-only accuracy across all four benchmarks:

- **Varying $k_{\text{attr}}$.**
  For each fixed $k_{\text{vis}} \in \\{2,4,8\\}$, increasing $k_{\text{attr}}$ from $2$ to $4$ consistently improves accuracy on all four datasets. This shows that retrieving a few more high-scoring attributes gives the model a richer textual description space without overwhelming it. When we further increase $k_{\text{attr}}$ from $4$ to $8$, the gains saturate or slightly regress, suggesting that additional lower-rank attributes mostly add redundancy rather than new information.

- **Varying $k_{\text{vis}}$.**
  For each fixed $k_{\text{attr}}$, $k_{\text{vis}} = 4$ is consistently as good as or slightly better than $k_{\text{vis}} = 2$, while $k_{\text{vis}} = 8$ does not bring further improvement and sometimes hurts performance. This pattern indicates that the model benefits from seeing a small set of complementary hard patches per retrieved concept, but aggressively increasing the number of patches exposes more noisy or less relevant regions and makes the system more sensitive to imperfect retrieval, without improving accuracy.

Overall, Figure 5(a) shows that a moderate choice $(k_{\text{attr}}, k_{\text{vis}}) = (4,4)$ achieves a good balance between performance and robustness. The curves are relatively flat around this region, which indicates that our system is not overly sensitive to small changes in these hyperparameters and remains robust even when some retrieved attributes or patches are suboptimal.

---

### Ablation on $k_{\text{patch}}$ (Figure 5(b))

Figure 5(b) evaluates the number of mined hard patches per image $k_{\text{patch}}$ used when building the visual index. We vary $k_{\text{patch}} \in \\{1,2,4,8\\}$ and report text-only accuracy for all four benchmarks.

The main trends are:

- Moving from $k_{\text{patch}} = 1$ to $k_{\text{patch}} = 2$ and $4$ consistently improves accuracy. A few complementary hard patches per image are enough to cover diverse stable traits of a concept, which makes retrieval more robust to pose and viewpoint changes.

- Increasing $k_{\text{patch}}$ beyond $4$ yields almost no further gain and sometimes a slight drop, while the index size and retrieval cost grow linearly with $k_{\text{patch}}$. This confirms that extra low-rank patches are largely redundant and can even introduce noise at training and retrieval time.

Based on this ablation, we use $k_{\text{patch}} = 4$ in all main experiments, which gives a good accuracy–efficiency trade-off.

---

### Robustness to retrieval errors

The patterns in Figure 5(a) and 5(b) also speak to robustness. When we deliberately increase $k_{\text{attr}}$, $k_{\text{vis}}$, or $k_{\text{patch}}$ beyond their best settings, we are effectively injecting more potentially noisy or less relevant elements into the retrieval pipeline. The fact that accuracy degrades only mildly, and remains stable over a broad range of moderate $k$ values, indicates that:

- The fused difficulty–relevance ranking in Algorithm 1 is effective at prioritizing truly informative attributes and patches.
- The decoder can tolerate a certain amount of noisy retrieved context, focusing attention on the most relevant external KV entries.

These new ablations in Figure 5(a) and 5(b) show that performance is stable in a reasonable range of $k$ values and that the default setting $(k_{\text{attr}}, k_{\text{vis}}, k_{\text{patch}}) = (4,4,4)$ offers a good accuracy–efficiency trade-off. We hope this clarifies the role of the top-$k$ design and addresses the reviewer’s concerns about hyperparameter sensitivity and robustness.

---

### Author Response · Authors · 2025-11-20
**Response to Reviewer tLpQ (W3 & W4)**

# Response to Reviewer tLpQ (W3 & W4)

---

## W3. Additional baselines (RePIC, PLVM, etc.)

> *Additional baselines (e.g., RePIC, PLVM, or other recent personalization methods) should be included to provide a more comprehensive evaluation.*

**Response.**

Thank you for this suggestion. Our initial submission selected a few representative methods from the two main families of personalization approaches, namely Yo’LLaVA and MC-LLaVA from the training-required line and RAP-LLaVA plus a prompt-concat variant on top of LLaVA-OneVision from the training-free, retrieval-based line.

In the revised version we have:

- **Added RePIC and PLVM as training-required baselines**, both reimplemented on the same LLaVA-OV backbone as Jarvis and other methods.
- **Updated Table 1 and Section 4.2** to report VQA, text-only QA, and recognition (Rec) for RePIC and PLVM under the same evidence and evaluation protocol.
- **Extended Section 4.3 and Figure 4** to compare latency and throughput for RePIC and PLVM as well, so that accuracy and efficiency can be viewed jointly.

These additions broaden the baseline coverage while keeping the backbone and decoding setup strictly aligned, and they show that Jarvis remains competitive or superior even when compared against these stronger training-required personalization models.

---

## W4. Beyond QA tasks and single-concept personalization

> *The experiments are confined to question-answering tasks. It remains unclear how the proposed framework would perform on other multimodal tasks such as image captioning, recognition, or cross-concept interactions. The work also focuses only on single-concept personalization without discussing scalability to multi-concept personalization.*

**Response.**

We appreciate this broader perspective. Our main experiments used QA as a unified probe for fine-grained personalization, but we agree that recognition and multi-concept behavior deserve explicit discussion.

1. **Recognition.**
   In the original submission, recognition ability was only indirectly tested through text-only QA. In the revision, we have **added an explicit recognition task for all methods** (Section 4.2). We report recognition accuracy (Rec) on four benchmarks in Table 1. Jarvis achieves the best Rec on Yo’LLaVA and essentially matches the strongest method on MC-LLaVA, which supports the claim that the external KV cache improves identity discrimination rather than only QA-style prompting.


2. **Multi-concept personalization.**
   The external KV design naturally extends to multi-concept queries by retrieving several concepts at once and concatenating their KV blocks into a shared prefix. In the revision we have added **a dedicated discussion in Appendix J**, where we describe how Jarvis can (i) retrieve the top few concepts relevant to a query, (ii) compose their caches into a single external prefix, and (iii) let the backbone reason about relations or comparisons between them. We also outline open directions such as more structured routing of concepts across heads or layers and adaptive selection of the number of concepts per query.

Overall, the new experiments and discussion highlight that the proposed external KV interface is not tied to a single QA template: it already supports recognition quantitatively, can be used for caption-style generation with the same backbone, and has a clear path toward scalable multi-concept personalization as discussed in Appendix J.

---

### Author Response · Authors · 2025-11-20
**Response to Reviewer tLpQ (W2)**

# Response to Reviewer tLpQ (W2)

---

## W2 — Clarifying novelty beyond KV-cache and retrieval-based personalization

> *The concepts of KV-cache and retrieval-based personalization have already been explored in prior works for both language and vision-language models. The paper should better clarify the technical differences between the proposed method and prior works, and ideally provide a fair empirical comparison to demonstrate its advantage.*

**Response.**

Thank you for raising this point. Our goal is not simply to reuse a KV-cache or to plug in a generic retriever, but to make concept-specific external KV states the main personalization mechanism for a frozen vision–language model, and to show that this design gives consistent gains on fine-grained identity benchmarks.

---

### Conceptual differences

Most related work falls into two groups.

1. **Prompt or token based retrieval.**
   These methods retrieve text or images and then rebuild a long prompt that is re-encoded at every turn. Even when a “memory” is used, it is stored as tokens. This keeps the backbone fixed, but context length and latency grow with every extra piece of evidence, and the retrieved evidence is usually at image or paragraph level.

2. **KV-cache aware language models.**
   Here the KV-cache is mainly an internal speed-up mechanism: it stores recent tokens within a session, is not organized per concept, and is not indexed or reused across sessions. Visual information is rarely integrated into the cache.

Our method differs in three ways.

- We **externalize precomputed concept KV**, not raw tokens. For each concept we run an offline prefill, turn its description into layerwise KV tensors on the frozen backbone, and store these tensors as reusable evidence. At inference time we never re-encode this information; we only attach the selected KV blocks to the current cache.

- We **jointly encode text and mined hard patches** into the same external KV memory. Each concept’s KV comes from fine-grained attributes and concept-only hard patches mined by the difficulty map. This puts attribute-level and region-level cues in the same space and makes the memory much more discriminative than using only captions or whole images.

- We **treat external KV as a persistent, retrieval-indexed concept memory**. Caches are keyed by concept id and can be reused across turns and sessions. Retrieval runs over a combined index built from attributes and hard patches, and only the few concepts selected by this index contribute KV to the decoder. This gives both efficiency (no repeated encoding) and robustness (small, well-chosen evidence sets).

To our knowledge, no prior work combines these three aspects: concept-level external KV precomputed offline, hard-patch visual fingerprints, and retrieval-driven attachment to a frozen VLM.

---

### Empirical comparison

We agree that these design choices must be justified empirically. In the revised experiments (Section 4.2) we compare, under a shared LLaVA-OV backbone:

- a **prompt only baseline** that concatenates concept text and images into a single long prompt without external KV or retrieval;
- **parameter based methods** including Yo'LLaVA, MC-LLaVA, PLVM and RePIC adapted to our backbone, where each concept has its own tokens or adapters;
- a **retrieval based baseline** similar in spirit to RAP-LLaVA, which stores concept exemplars and injects them via prompts.

All methods use the same evidence, datasets and evaluation protocol. Our approach is training free on top of LLaVA-OV; baselines follow their own training pipelines.

Across Yo'LLaVA, MC-LLaVA and the harder Yo'LLaVA++ and MC-LLaVA++ benchmarks, Table 1 shows that the external KV approach achieves the best accuracy in both text-only QA and VQA, with the largest margins on the ++ datasets where fine-grained identity cues matter most. This indicates that using concept-specific external KV populated with hard patches is not only different in form, but also brings clear practical benefits over existing prompt-based and KV-aware personalization methods.

---

### Author Response · Authors · 2025-11-20
**Response to Reviewer tLpQ (W1)**

# Response to Reviewer tLpQ (W1)

---

## W1 — Clarifying retrieval similarity, dataset construction, and QA formats

> *Several implementation details remain insufficiently explained. For example, it is unclear how textual similarity is computed for retrieval, how the proposed Yo'LLaVA++ and MC-LLaVA++ datasets are constructed, and what the question formats in these datasets look like. The authors should provide more detailed descriptions in the main text or appendices to improve clarity and reproducibility.*

**Response.**

We thank the reviewer for carefully pointing out the missing implementation details and fully agree that these are important for clarity and reproducibility. In the revised manuscript we have made the following concrete changes.

1. **Textual similarity for retrieval.**
We follow Appendix D and build a text-only semantic index over concepts. For each concept $c$ we first concatenate its name, caption, and top-$K$ fingerprint attributes into a short document $D(c)$. We then feed $D(c)$ into the base language model embedding, mean-pool over tokens, and apply $\ell_2$ normalization to obtain a single vector per concept:

$$
e_c = \frac{\mathrm{LMEmb}(D(c))}{\left\|\mathrm{LMEmb}(D(c))\right\|_2}.
$$

At inference time, a user query $q$ is encoded in exactly the same way:

$$
e_q = \frac{\mathrm{LMEmb}(q)}{\left\|\mathrm{LMEmb}(q)\right\|_2}.
$$

The textual similarity between the query and concept $c$ is then defined as the cosine score:

$$
s_{\text{text}}(c \mid q) = e_q^\top e_c .
$$

We rank all concepts by $s_{\text{text}}(c \mid q)$ and take the top-$m$ concepts as the retrieved set used for external KV selection, unless the query already contains an explicit concept placeholder token, in which case we map it directly as described in Appendix D.


2. **Construction of Yo'LLaVA++ and MC-LLaVA++.**
We have added a dedicated Appendix E titled “Construction of Yo'LLaVA++ and MC-LLaVA++”, which presents the full pipeline within a unified conceptual framework. For each concept, we describe the required inputs, including the concept ID, the associated images, and the optional fingerprint attributes and hard-region metadata. We then explain how a compact set of visual references is selected, how trait anchors and hard-region cues are derived, how instruction prompts are constructed, and how the model outputs are parsed and post-processed. The appendix follows the exact logic implemented in our generation script and is organized to provide a clear and coherent end-to-end overview of the system.

3. **Question formats and prompts.**
   The exact formats of the generated questions are now documented in Appendix E as well. We include the full prompt templates for both the concept-level setting and the per-image setting, and we spell out the JSON schema used for each question, with fields `"question"`, `"option"` (containing `"A"` and `"B"`), and `"correct_answer"`. We also describe how these question objects are organized in the Yo'LLaVA++ and MC-LLaVA++ files. This makes the data format fully explicit and aligned with the implementation.

Taken together, these additions make the retrieval mechanism and dataset construction fully reproducible: Appendix D details the external KV construction and decoding procedure, Appendix E gives the unified pipeline and prompt templates for building Yo'LLaVA++ and MC-LLaVA++, and Appendix K further provides several concrete examples from both datasets.

---

### Author Response · Authors · 2025-11-20
**Response to Reviewer tLpQ (Q2 & Q3)**

# Response to Reviewer tLpQ (Q2 & Q3)

---

## Q2. Clarifying the RAP-LLaVA backbone

> *Regarding the RAP-LLaVA baseline, is the model used in the experiments the officially released version, or was it retrained using the LLaVA-OneVision backbone for fair comparison? Clarifying this is important for assessing experimental consistency.*

**Response.**

Thank you for raising this point about experimental consistency.
In our experiments, **RAP-LLaVA is not the original released checkpoint**; we **retrained RAP-LLaVA on top of the same LLaVA-OneVision (LLaVA-OV) backbone** used by Jarvis.

Concretely:

- We keep the **vision tower, language backbone, and tokenizer identical** to LLaVA-OV in all training-free and training-required methods.
- For RAP-LLaVA, we **follow the authors’ training recipe and loss objectives**, but replace the original backbone with LLaVA-OV and re-run training on the same personalization datasets as in the original paper.
- For the other training-required baselines (Yo’LLaVA, MC-LLaVA, RePIC, PLVM), we similarly adopt **LLaVA-OV as the shared backbone** and either (i) replicate the official training pipeline with this backbone, or (ii) adapt their released implementation with minimal changes so that only the backbone is swapped while the personalization head, loss, and data remain the same.

This setup ensures that **all methods differ only in their personalization mechanism** (prompt concatenation, retrieval, adapters, or external KV) rather than in backbone capacity or vision-language alignment quality. We will clarify this in the revision by explicitly stating in Section 4 and in the caption of Table 1 that all baselines are evaluated on a unified LLaVA-OV backbone and that RAP-LLaVA is retrained accordingly.

---

## Q3. Latency and throughput comparison

> *In Figure 4, why is MC-LLaVA not included in the latency and throughput comparison?*

**Response.**

Thank you for pointing this out; you are right that the initial version of Figure 4 only compared Jarvis with a subset of baselines. Our original intention in Section 4.3 was to highlight two complementary contrasts: (i) Jarvis versus a **training-free retrieval-style method** (RAP-LLaVA), and (ii) Jarvis versus a **training-based token-personalization method** (Yo’LLaVA).
Under this focus, we omitted MC-LLaVA to keep the plot compact.

We agree that a complete view including all personalization baselines is more informative. In the revised version, we have therefore:

- **Extended the latency and throughput experiments** to include MC-LLaVA, RePIC, and PLVM in addition to Jarvis, Prompt-concat, Yo’LLaVA, and RAP-LLaVA, all on the same hardware and decoding setup.
- **Updated Figure 4** to show Jarvis versus every baseline model, and
- **Revised Section 4.3** to discuss the expanded results and to make clear that Jarvis maintains the lowest per-turn latency and highest QPS even when compared against all training-required methods, not only Yo’LLaVA.

These additions should address the concern and provide a more complete empirical picture of Jarvis’s efficiency relative to both retrieval-based and parameter-based personalization baselines.

---

### Author Response · Authors · 2025-11-20
**Response to Reviewer tLpQ (Q1)**

# Response to Reviewer tLpQ (Q1)

---

## Q1. Handling queries that do not explicitly name the target concept

> *The evidence retrieval process appears to rely primarily on textual similarity. How does the system handle cases where the target concept is not explicitly mentioned in the query text?*

**Response.**

Thank you for raising this point. Our retrieval mechanism is designed to handle both cases where the concept is named explicitly in the query and cases where it is only described implicitly. Appendix D (“External KV Construction and Decoding”) now makes this explicit.

---

### Two retrieval modes: explicit placeholders vs implicit descriptions

Our system operates in two modes, depending on the query form:

1. **Explicit placeholder in the query.**
   When the query text contains a special placeholder, such as `⟨cat-cup⟩` or `⟨Eru⟩`, we do *not* rely on similarity search at all. Instead, we directly map the placeholder to its known concept id and set
   $S(q, I)$ to that concept (or a small fixed set if multiple placeholders are present).
   In this case retrieval is purely symbolic and does not depend on the wording of the rest of the question.

2. **No explicit concept name in the query.**
   When the query does not contain any placeholder, we use *semantic* retrieval rather than string matching. For each concept $c$, we build a short document
   $$
   D(c) = \text{concept name} + \text{caption} + \text{top-}K \text{ fingerprint attributes},
   $$
   as described in Appendix D, Eq. (16). We encode $D(c)$ with the base model’s token embedding and mean-pool across tokens to obtain a normalized vector $e_c$ (Eq. (17)).
   At inference time, we encode the query $q$ in exactly the same way to obtain $e_q$ (Eq. (18)) and compute the cosine similarity
   $$
   s_{\mathrm{text}}(c \mid q) = e_q^\top e_c,
   $$
   as in Eq. (19). We then take the top-$m$ concepts (typically $m \le 2$) according to $s_{\mathrm{text}}(c \mid q)$ as $S(q, I)$ (Eq. (20)).

Crucially, this semantic encoding allows us to retrieve the correct concept even when the user does not mention its name. As long as the question contains distinctive textual cues (for example “the dog with one blue eye and a black patch on the left ear”), the embedding $e_q$ aligns with the concept’s description and fingerprint attributes stored in $D(c)$, so the correct concept is ranked among the top candidates.

As a result:

- When the concept is explicitly named (via a placeholder), we bypass retrieval and directly select the correct concept;
- When the concept is only described, the query and the concept’s metadata share a common attribute vocabulary, making the semantic similarity $s_{\mathrm{text}}(c \mid q)$ a reliable signal.

We have clarified this two-mode behavior and the role of $D(c)$, $e_c$, and $s_{\mathrm{text}}(c \mid q)$ in Appendix D, so that it is clear how the system remains effective even when the target concept is not explicitly named in the query text.

---

### Author Response · Authors · 2025-11-20
**Response to Reviewer DAdL (W4)**

# Response to Reviewer DAdL (W4)

---

## W4 — Scenario 2 in Figure 1

> *In scenario 2 of Figure 1, the paper claims that Jarvis can clearly highlight the difference. However, from the answer, we do not know what the usual clothes [object Object] are. The difference between RAP-LLaVA and Jarvis only comes from "Different from usual casual attire"; however, it does not indicate that RAP-LLaVA cannot follow the instruction. Maybe with a different seed, the response of RAP-LLaVA includes the term "Different from usual casual attire".*

**Response.**

Thank you for pointing out this ambiguity. We agree that the original version of Scenario 2 mixed visual evidence with an underspecified phrase (“usual casual attire”), which is not directly recoverable from the displayed images and leaves room for interpretation about what counts as “usual.” This also makes it harder to argue that RAP-LLaVA truly fails to follow the instruction.

In the revised manuscript we have **replaced Scenario 2 with a clearer, fully image-grounded setup**, shown in the new Figure 1:

- The question is now
  *“What is \<viruss\> doing in the left photo, and how is his outfit different from the other two photos?”*
  so the required comparison is **explicitly between the three provided photos**, with no reference to “usual” style.
- Yo’LLaVA answers only *“\<viruss\> is taking a selfie,”* ignoring the comparison request.
- RAP-LLaVA describes the images but still does not articulate a concrete contrast between the left photo and the others.
- **Jarvis** answers:
  *“In the left photo \<viruss\> is taking a selfie in a dark blue suit with a black shirt and a lapel pin. In the middle and right photos he is not holding a phone and wears lighter, more casual clothes.”*
  This response directly contrasts **phone vs. no phone** and **dark formal suit vs. lighter casual clothes**, which are both visible differences in the three images and do not rely on any external, unstated notion.

To remove randomness in this qualitative comparison, all three methods in Figure 1 are decoded with **temperature 0** and a fixed decoding configuration, so the illustrated outputs are deterministic under our setup rather than cherry-picked from multiple samples. We also emphasize that this figure is intended as a qualitative illustration; the main evidence for Jarvis’s advantages comes from the quantitative results and ablations in Sections 4.2 and 4.3.

---

### Author Response · Authors · 2025-11-20
**Response to Reviewer DAdL (W3)**

# Response to Reviewer DAdL (W3)

---

## W3 — Single vs multi-concept personalization

> *Restricted only to single concept personalization: Jarvis only addresses the problem of Single concept personalization. However, it is running on LLaVA-OV, which can accept multiple input images to the input, thus having the capability of multiple-concept personalization in the architecture of LLaVA-OV. In addition, conducting only single-concept personalization is not convincing, since most of the dataset for single-concept personalization contains only a single object of that concept, thus leading to the scenario that the LVLM can respond to the query input, while not being aware of what the user is referring to.*

**Response.**

Thank you for raising this important concern. We address it in three parts:

---

### Why we focus on single-concept personalization in this work

Our goal in this paper is to study **how concept-specific information should be represented and reused** inside a frozen VLM, not to exhaustively cover all possible personalization scenarios at once. Single-concept personalization is a natural first step for this goal because it isolates a clear question:

> Given a large vision–language backbone, can we build an external concept memory that reliably captures *who* the subject is and reuse it across many queries without retraining?

Within this controlled setting, Jarvis departs from prior work in two ways:

- Personalized information is stored as **precomputed external KV caches** (built from text attributes and mined hard patches) rather than as extra tokens or trainable adapters.
- At inference time, the model **reuses these concept KV caches directly**, instead of re-encoding all reference images and descriptions every turn.

The single-concept setup allows us to compare this design fairly against prompt-based and training-based baselines on the same backbone, and to measure its effect on fine-grained identity reasoning without confounding factors from entity disambiguation.

---

### Why the single-concept benchmarks are still challenging

We agree that “one concept per image” can be misleading if interpreted as “one salient object, no ambiguity”. However, the datasets we use are **not** trivial in this sense:

- Yo’LLaVA, MC-LLaVA and especially their **++ variants** contain many **visually similar concepts** (for example, different pets or people with similar poses, clothing styles, and backgrounds). The model must rely on **subtle, identity-bearing cues** such as local patterns, small accessories, and fine geometry rather than simply detecting the coarse class.

- The **text-only QA** and **recognition (Rec)** settings explicitly remove the test-time image or require identifying the correct concept among distractors. In these cases, the model cannot just “look at the single object in the image”; it must **link the query to the right concept memory** and distinguish it from other candidates.

- Our results show large gains precisely on these hard settings: Jarvis achieves substantial improvements on Yo’LLaVA++ and MC-LLaVA++ text-only QA and competitive or best recognition performance, even when competing against training-based methods. This indicates that the model is not merely responding to “something in the image” but is using the external KV memory to resolve concept identity.


---

### How Jarvis extends to multi-concept personalization

We fully agree that multi-concept queries are central to real assistants. Jarvis is not limited to a single entity; the same external-KV design naturally scales to several concepts in one turn.

At build time, we keep a repository of per-concept external KV caches
$\mathrm{KV}^{(c)}$ constructed from text attributes and mined hard patches. At inference time, instead of selecting only one concept, the retriever can return the top $K_{\text{concept}}$ relevant concepts. Their caches are then concatenated into a shared external prefix, followed by the current turn’s KV; the backbone attends over these segments and can compare or relate concepts within a single decoding pass.

Scalability is therefore controlled by $K_{\text{concept}}$,
which can be kept small for comparison-style queries. All heavy work (hard-patch mining and KV construction) remains per-concept and offline, while per-turn cost is limited to retrieval, cache composition, and one pass of decoding. In this sense, complexity grows with the number of retrieved concepts, not with the total number of personalized concepts stored in the system. We describe this extension and related design choices in more detail in Appendix J.

---

### Author Response · Authors · 2025-11-20
**Response to Reviewer DAdL (Q4-2)**

# Response to Reviewer DAdL (Q4-2)

---

## Q4-2. The top-k elements in Algorithm 1

> *Why do we need to select the top-k elements in Algorithm 1? Do we conduct an ablation study for k?*


**Response.**

Thank you for pointing this out. The choice of a top-k operation in Algorithm 1 is motivated by both efficiency and representation quality, and we have now added an explicit ablation over this parameter in the revised manuscript (see Figure 5(b)).

---

### Why a top-k step is needed

In Algorithm 1 we score many candidate windows on each training image by the fused map $C_m^w$ and obtain a score $s_m(u,v)$ for each window. In practice, a single concept image can easily yield tens of candidate windows. Keeping all of them would be possible, but it introduces several concrete issues.

1. **Controlling computation and memory.**
   Every mined window is turned into a hard patch that enters the visual index. If we keep all candidates, the number of patches per concept grows almost linearly with the number of images, which inflates the index, increases similarity computations at retrieval time, and lengthens the external key–value prefix used during decoding. This goes against our goal of keeping the retrieval module lightweight and easily pluggable.

2. **Filtering redundancy and noise.**
   High-score windows tend to cluster on a few key regions that are both difficult for the diffusion model to reconstruct and strongly aligned with the concept text (for example, eye color, muzzle markings, specific logo areas). Lower-score windows are often either marginal parts of the subject or noisy responses of $C_m^w$. If we retain all windows, the visual index becomes dominated by many nearly redundant or weakly relevant patches, which dilutes the effect of the truly discriminative ones.


For these reasons, the top-k step is a controlled compression mechanism: it preserves the most informative concept-specific patches and discards long-tail regions that are either redundant or noisy.

---

### Ablation over $k_{\text{patch}}$ (Figure 5(b))

We agree that the effect of $k_{patch}$ is important to report. In the revised version we added an explicit ablation in Figure 5(b), where we vary the number of mined hard patches per image, denoted \(k_{\text{patch}}\), when building the visual index. We evaluate \(k_{\text{patch}} \in \{1, 2, 4, 8\}\) on all four benchmarks.

The main observations are:

- **Moderate $k_{\text{patch}}$ clearly helps.**
  Moving from 1 to 2–4 patches per image consistently improves text-only accuracy across all datasets. A few complementary hard patches are enough to capture diverse, stable traits of a concept and make retrieval more robust.

- **Large $k_{\text{patch}}$ brings little benefit.**
  Increasing $k_{\text{patch}}$ beyond 4 yields almost no further gains and sometimes a slight drop, while the index size and retrieval cost grow linearly. This indicates that the extra low-ranked patches are mostly redundant or noisy.


Based on this ablation, we set $k_{\text{patch}} = 4$ in all main experiments, which provides a good trade-off between accuracy and efficiency. We have updated Section 4.4.1 to explicitly connect Algorithm 1 to this ablation and to make clear that the top-k operation is not tuned aggressively but is stable over a small range of values.

---

### Author Response · Authors · 2025-11-20
**Response to Reviewer DAdL (Q4-1)**

# Response to Reviewer DAdL (Q4-1)


---

## Q4-1 — How are concept patches used at test time?

> *Do we add patch $P^{(c)}$ for each concept to the LLaVA-OV in test time, and how do we add that?*

**Response.**
We appreciate this question, since it directly concerns the computational cost and the practical interface with LLaVA-OV at inference time. Our design is intentionally conservative: (i) we never activate all concepts at once, and (ii) hard patches are only used to refine textual evidence, not as additional image inputs to the backbone VLM.

---

### Do we attach patches for every concept at test time?

No. At test time we only consider a small number of concepts that are relevant to the current query, rather than injecting patches for all concepts.

Concretely, given a user query (and optionally an image), we first perform semantic retrieval over all concepts and select a small candidate set
$$
\mathcal{S}(q,I) = \\{c_1,\dots,c_m\\},
$$
where the size $m$ is very small (for example one or two concepts in most cases).

Only concepts in $\mathcal{S}(q,I)$ participate in the subsequent external KV assembly and patch-based reasoning. All other concepts remain inactive: they do not contribute any KV cache entries and do not incur inference cost.

---

### For selected concepts, how are patches $P^{(c)}$ actually used?

For each concept $c \in \mathcal{S}(q,I)$, the hard patches $P^{(c)}$ are not fed to LLaVA-OV as image patches. Instead, they serve as a tool to identify which fine-grained visual traits should be verbalized as evidence.

In more detail:

1. **Offline: patches build a concept-level visual index.**
   During hard patch mining, we extract a set of concept-only patches $P^{(c)}$ for each concept and encode each patch with a CLIP-style image encoder. These patch embeddings form a visual index for concept $c$. This index is only used to measure which local regions are most relevant to a given query; the patches themselves are not used as additional visual tokens in LLaVA-OV.

2. **Online: patches select evidence, which is then converted into text.**
   At inference time, given the current question $q$:

   - We encode $q$ with a text encoder and compute similarity between the query embedding and the patch embeddings in $P^{(c)}$, selecting a few hard patches that are most relevant to the question (for example a left ear region, a tail base region, or a distinctive marking).
   - From these selected patches, we derive concise textual hints that describe the corresponding fine-grained traits (for example “left ear region shows darker, edged fur”). These hints are appended to the concept’s metadata and then linearized into the short prefix $\tau^{(c)}$ that is used to build the external KV cache.

In other words, patches $P^{(c)}$ at test time are used to decide *which fine-grained, subject-specific attributes to mention in text*. They influence the model indirectly through the external KV constructed from these textual hints; they are not inserted as extra image tokens into LLaVA-OV.

---

### Summary

To summarize:

- We do **not** attach patches for all concepts at test time. Only the small retrieved set $\mathcal{S}(q,I)$ is activated per query.
- For each activated concept, the hard patches $P^{(c)}$ are used to drive the selection of fine-grained textual evidence, which is then injected via external KV. The backbone VLM still sees a single image input and a text prefix, without a proliferation of image patches.

We have revised Appendix D to describe this behavior in detail and to clarify that we do not directly feed a large number of image patches into LLaVA-OV at inference time.

---

### Author Response · Authors · 2025-11-20
**Response to Reviewer DAdL (Q3)**

# Response to Reviewer DAdL (Q3)

---

## Q3 — On the derivation details in Appendix D

> *In Appendix D, what is the linearization of $T^{(c)}$ into short prefix $\tau^{(c)}$ (Line 657), and how do we retrieve the small concept set $S(q, I)$ in Line 668?*

**Response.**
We thank the reviewer for raising this detailed question. The issue concerns two essential components of our pipeline:
(1) how the structured concept metadata $T^{(c)}$ is transformed into the short prefix $\tau^{(c)}$ used for prefill, and
(2) how the inference procedure retrieves the small concept set $S(q,I)$.
We have clarified these points more explicitly in the revised Appendix D, and summarize the key ideas here.

---

### From $T^{(c)}$ to the short prefix $\tau^{(c)}$

Each concept is stored as a structured metadata bundle,

$$
T^{(c)} = \\{\mathrm{concept},\ \mathrm{category},\ \mathrm{caption},\ \mathrm{fingerprint\\_attributes}\\}.
$$



The conversion from $T^{(c)}$ to the short prefix $\tau^{(c)}$ is entirely rule-based and parameter-free.

1. **Filtering and selecting attributes.**
   We remove attribute phrases in `fingerprint_attributes` that merely describe backgrounds or logos (e.g., “often on a bed”), retaining only those that reflect genuine subject-specific details. We keep at most $K$ attributes (e.g., $K=8$) so that the prefix remains concise and semantically dense.

2. **Composing a fixed-template natural-language prefix.**
   The fields `concept`, `category`, `caption`, and the selected attributes are assembled into a short textual description using a fixed template, for example:

$$
\mathrm{Subject:\ \langle concept\rangle.\ Category:\ \langle category\rangle.\ Description:\ \langle caption\rangle.\ Distinctive\ traits:\ attr\_1;\ \ldots;\ attr\_K.}
$$


   Each attribute $attr_k$ is taken verbatim from the filtered `fingerprint_attributes`. Since the template is fixed, a given $T^{(c)}$ always yields an identical textual prefix, introducing no learnable components.

3. **Tokenization and length truncation.**
   The resulting prefix string is fed into the tokenizer of the base LLM $f_\theta$. If the sequence exceeds the maximum length $L_{\max}$, we keep the first $L_{\max}$ tokens. The final token sequence constitutes $\tau^{(c)}$.

We then perform a one-time prefill for each concept:

$$
(K^c_{1:L},\ V^c_{1:L})
= \mathrm{Prefill}(f_{\theta},\ \tau^{(c)}).
$$

which produces the persistent KV cache $KV^{(c)}$, reused across all future dialogs while $f_\theta$ remains frozen.

---

### Retrieving the small concept set $S(q, I)$

The construction of $S(q,I)$ is driven entirely by the query text $q$. The input image $I$ does not affect concept selection.

1. **Offline construction of a semantic index.**
   For each concept $c$, we first assemble a short semantic document

    $$
    D^{(c)} = \mathrm{concept} + \mathrm{caption} + \mathrm{top}\text{-}K\ \mathrm{fingerprint\\_attributes}.
    $$


   We encode $D^{(c)}$ using the base model’s input embedding layer, perform mean pooling over tokens, and normalize the result to obtain the semantic embedding

   $$
   \mathbf{e}^{\mathrm{sem}}_c = \operatorname{LMEmb}(D^{(c)}).
   $$

   This provides a fixed vector representation for each concept for similarity-based retrieval.

2. **Online encoding of the query and similarity scoring.**
   The user query $q$ is encoded in the same way:

   $$
   \mathbf{e}^{\mathrm{sem}}_q = \operatorname{LMEmb}(q),
   $$

   followed by cosine similarity computation:

   $$
   s_{\mathrm{text}}(c \mid q)
   = \langle \mathbf{e}^{\mathrm{sem}}_q,\ \mathbf{e}^{\mathrm{sem}}_c \rangle.
   $$

   If $q$ explicitly contains placeholders (such as `<cat-cup>`), we directly map them to the corresponding concept identifiers.
   Otherwise, we rank all concepts by $s_{\mathrm{text}}(c \mid q)$ and select the top $m$ (with $m$ being small, usually $m \le 2$):

    $$
    \mathcal{S}(q,I) = \{c_1,\dots,c_m\}
    = \operatorname{Top}\text{-}m_{c}\ s_{\mathrm{text}}(c \mid q).
    $$

---

In summary, the linearization $T^{(c)} \rightarrow \tau^{(c)}$ is achieved via a fixed template and token-length truncation, without introducing any learnable parameters. The set $S(q,I)$ is obtained through a deterministic semantic-similarity retrieval mechanism. These clarifications have been added to the revised manuscript to better connect the derivation in Appendix D with the actual implementation.

---

### Author Response · Authors · 2025-11-20
**Response to Reviewer DAdL (W2 & Q2)**

# Response to Reviewer DAdL (W2 & Q2)


---

## W2 & Q2 — On the difficulty map $C_m$ and the motivation of several update steps in Algorithm 1

> *The motivation behind the algorithm is not clearly explained: For example, Algorithm 1 contains a term called the difficulty map $C_m$, but it is not clearly explained in the main paper why taking the Diffusion Inversion can result in a difficulty map. Other terms such as background negative $\mathcal{B}$ are not clearly explained in Algorithm 1. Also, what is the motivation behind the update in Lines 6, 10, and 11 in Algorithm 1?*

**Response.**
We fully agree that the motivation behind the algorithm should be explained in a more intuitive way. In the revised version, we add a higher-level discussion in Appendix B.2–B.4. The core idea can be summarized as follows: **we use the unconditional prior of a diffusion model to characterize “where the model finds it hard to generate”, then use text relevance to filter out concept-irrelevant complex backgrounds, and finally apply subject-coverage and region averaging to select local patches that are both “difficult and concept-relevant”.** Concretely:

1. **Why does diffusion inversion yield the difficulty map $C_m$?**
   * We regard the pretrained text-to-image diffusion model as a strong natural-image prior. For common background regions (walls, bed sheets, floors, etc.), the unconditional diffusion model is already very good at generation. During the “inversion–regeneration” process, these regions require only very small local updates to be reconstructed.
   * In contrast, fine-grained details that encode individual identity (e.g., specific patterns around a pet’s eyes or a small notch in a logo) are parts where the prior is less familiar, and therefore they require larger local updates to match the real image.
   * In implementation, we perform diffusion inversion on the real image $I_m$ to obtain a latent trajectory $\\{z_t^{(m)}\\}_{t=1}^T$, and then run forward denoising along this trajectory in the unconditional setting. We accumulate, at each spatial location, the magnitude of local changes throughout the denoising process. Locations with larger and more persistent changes are harder to explain by the unconditional prior and thus receive higher values in $C_m(x,y)$.
   * Consequently, $C_m$ can be viewed as a heatmap of regions that the diffusion model finds hardest to reconstruct without text guidance, typically aligning with instance-specific high-frequency textures and fine structures.

2. **Why do we multiply $C_m$ with the text relevance $R_m$ in Line 6?**
   * If we rely only on $C_m$, all “hard-to-generate” content will be treated as candidates, including visually complex but concept-irrelevant backgrounds (e.g., carpet textures). If we rely only on $R_m$, the selection will be dominated by “semantically relevant but very common” regions (e.g., large areas of fur).
   * Therefore, we fuse them as
     $$
     C_m^w = \mathrm{normalize}\big(C_m \cdot R_m^\gamma\big)\odot M_m,
     $$
     so that only pixels that are simultaneously difficult for the unconditional diffusion model to reconstruct (high $C_m$) and highly relevant to the concept text while not belonging to the background (high $R_m$) are emphasized in $C_m^w$. The exponent $\gamma$ controls the weight of text relevance; in our experiments we use a fixed value.

3. **Why do we introduce the coverage ratio $\kappa$ in Line 10?**
   * In Line 10, we compute
     $$
     \kappa = \frac{|B_m(u,v)\cap M_m|}{|B_m(u,v)|},
     $$
     i.e., the fraction of subject pixels inside a candidate patch, and we require $\kappa \ge \eta$.
   * Even though $C_m^w$ has been cropped by the mask $M_m$, patches near the boundary may still contain a large proportion of background pixels. Our goal is that the patches used to build the External KV should encode **the subject’s internal visual fingerprint** as much as possible, rather than a mixture of subject and large background regions.
   * The threshold $\eta$ thus serves as a simple but crucial subject-coverage constraint: only patches with sufficiently high subject coverage are eligible for ranking and global top-$k$ selection.

4. **Why do we use region averaging in Line 11 instead of max pooling?**
   * In Line 11, we define the patch score as
     $$
     s_m(u,v) = \frac{1}{|B_m(u,v)|} \sum_{(x,y)\in B_m(u,v)} C_m^w(x,y).
     $$
   * We choose averaging mainly for two reasons:
     (i) **Stability.** Averaging suppresses the effect of isolated high-value pixels or small fluctuations near patch boundaries, leading to more stable scores;
     (ii) **Comparability.** Under a fixed patch size, averaged scores are more comparable across different images and locations, which facilitates global ranking and top-$k$ selection.

We hope these additions better address your concerns about the insufficient explanation of the motivation behind Algorithm 1.

---

### Author Response · Authors · 2025-11-20
**Response to Reviewer DAdL (W1 & Q1)**

# Response to Reviewer DAdL (W1 & Q1)

We thank **Reviewer DAdL** for carefully reading our paper and for pointing out that some terms in Algorithm 1 (such as the background negative set $\mathcal{B}$ and the background map $R_{m,b}^{-}$) were not clearly defined. This helped us realize that the presentation of hard patch mining was not sufficiently self-contained, and we have clarified the notation in Appendix B.

---

## W1 & Q1 — On the clarity of Algorithm 1

> *What is the definition of some terms: background negative $\mathcal{B}$, background map $R_{m,b}^{-}$ in Algorithm 1? In this case, the paper should give a clearer definition of these terms, because it is hard to follow and understand the patch mining presented in Algorithm 1.*

**Response.**
We appreciate your observation regarding the insufficient clarity of several terms in Algorithm 1. Our intention is to ensure that each symbol is immediately understandable to readers in terms of its definition and functional role within the hard-patch mining pipeline. In the revised manuscript, we now provide more explicit and concise explanations, together with a consolidated reference table in Appendix B.

1. **Candidate patch $B_m(u,v)$**
   The term $B_m(u,v)$ denotes a candidate patch extracted from the $m$-th image $I_m$. It corresponds to the pixel set of the $(u,v)$-th window obtained from a regular grid or sliding-window partition at a unified resolution (for example, $512\times512$). This patch is used to compute the subject-mask coverage within the region and to aggregate pixel-level difficulty–relevance values via regional averaging, producing a stable patch score $s_m(u,v)$ that reduces the influence of individual pixel noise.

2. **Positive OpenCLIP relevance map $R_m^{+}$**
   The map $R_m^{+} \in [0,1]^{H\times W}$ is generated by applying OpenCLIP to image $I_m$ under the concept text $T$, using a Grad-CAM-inspired method to obtain pixel-level semantic relevance. Each pixel value indicates how well the local appearance aligns with the target concept. This serves as a semantic prior and is later fused with the difficulty map $C_m$.

3. **Background negative set $\mathcal{B}$ and background relevance maps $R_{m,b}^{-}$**
   The set $\mathcal{B}$ contains textual descriptions of backgrounds such as “bed sheets”, “living room”, and “ceiling”. These descriptions frequently appear in personalization datasets but do not convey identity-bearing information. For each $b \in \mathcal{B}$, we compute a background relevance map $R_{m,b}^{-} \in [0,1]^{H\times W}$ by running OpenCLIP on $I_m$ conditioned on $b$.
   These background maps are used to construct a background-suppressed relevance map:

   $$
   R_m = \mathrm{normalize}\Big(\mathrm{ReLU}\big(R_m^{+} - \max_{b\in\mathcal{B}} R_{m,b}^{-}\big)\Big),
   $$

   which explicitly down-weights activations that can be equally well explained by background descriptions and retains only the pixels that are significantly more aligned with the concept text $T$.

4. **Additional explanations in the appendix**
   Appendix B now provides a step-by-step explanation of Algorithm 1 in execution order, describing the motivation behind each computation and its correspondence with the pseudocode. Table 3 consolidates all symbols and hyperparameters used in Algorithm 1, along with their meanings and experimental settings, which improves clarity and reproducibility.

These revisions ensure that readers can clearly understand what each symbol represents, how it is computed, and what role it plays in the mining pipeline, thereby addressing your concern that the original description of patch mining was difficult to follow.

---

### Author Response · Authors · 2025-11-20
**Response to Reviewer JB1P (W2)**

# Response to Reviewer JB1P (W2)

---

## W2. Dynamic concepts and updating external KV caches

> *The paper presents evidence construction as a one-time, offline process. However, user-specific concepts are often dynamic; users provide new images or information over time. The paper does not discuss a mechanism for efficiently updating or augmenting the KV-Cache for an existing concept without re-running the entire heavy offline pipeline. This is a practical limitation for long-term personalization.*

**Response.**

Thank you for raising this practical point. Conceptually, Jarvis separates **what evidence we extract** from user data and **how that evidence is injected** into the model via external KV. The KV injection stage only expects a compact text bundle and a small set of hard patches per concept; it does not depend on how these were obtained. This is deliberate, so that evidence can be refreshed or replaced without changing the core mechanism.

In a long-term setting, an existing concept does not need to be rebuilt from scratch. When new images or information arrive, one can run the hard-patch miner and attribute summarizer **only on the new data**, merge the resulting attributes and patches with a bounded subset of old ones, and then recompute the external cache $\mathrm{KV}^{(c)}$ from this updated evidence. The cost is proportional to the incremental update, not to the entire history. This can be done periodically in the background (for example, nightly) or on demand when a significant change is detected.

More broadly, because Jarvis only cares about the final text+patch interface, the upstream concept extractor can be any system: log analysis tools, external models, or manual curation. As users evolve, these tools can maintain and refine concept evidence, and Jarvis can immediately turn the updated summaries into new KV caches without additional fine-tuning. The offline stage is intended as a pluggable component, not a one-time immutable step.

---

### Author Response · Authors · 2025-11-20
**Response to Reviewer JB1P (W1)**

# Response to Reviewer JB1P (W1)

---

## W1. Multi-concept queries and scalability of KV-based personalization

> *The experiments primarily focus on single-concept personalization within a given session. A key challenge for a real-world AI assistant is handling queries that involve interactions between multiple personalized concepts (e.g., "my dog," "my daughter," "my car"). It is unclear how the proposed KV-Cache retrieval and concatenation would scale in complexity and performance when a query ambiguously references several distinct entities.*

**Response.**

Thank you for highlighting this important use case. We fully agree that realistic assistants must handle queries that involve several personalized entities and potentially ambiguous references. Our framework was designed with this extension in mind, and we summarize how Jarvis generalizes beyond the single-concept setting.

---

### From single-concept to multi-concept retrieval

In the main experiments, we intentionally restrict the setup to a single resolved concept in order to evaluate identity sensitivity under controlled conditions. However, the pipeline is already concept-indexed:

- For each concept $c$, we precompute and store a compact text bundle, mined hard patches, and the corresponding external KV cache $\mathrm{KV}^{(c)}$.
- At inference time, given a query (and optional image), we score all concepts and select the most relevant one.

For multi-concept queries such as *“Compare my dog and my daughter in these photos”* or *“Which outfit matches my car’s color best?”*, the retrieval stage simply returns the **top $K_{\text{concept}}$** concepts instead of a single one. Rather than changing the backbone or the KV format, we only change the granularity of the retrieval result.

In practice, $K_{\text{concept}}$ can be kept small (e.g., two or three for comparison-style questions), so that we only bring in concept caches that are strongly supported by the joint text-image similarity score.

---

### Composing KV caches across concepts

Once retrieval identifies a small set of candidate concepts, Jarvis composes their external caches by concatenation along the sequence dimension:

- For each selected concept $c$, we load $\mathrm{KV}^{(c)}\$, which already encodes its textual attributes and hard patches.
- We concatenate these blocks into a shared external prefix, then append the current turn’s cache that encodes the user query and image.

This is a direct generalization of the single-concept case, where $K_{\text{concept}} = 1$. The complexity grows roughly linearly with the number of retrieved concepts, and remains manageable as long as we keep $K_{\text{concept}}$ small. Crucially, we never re-encode concept images or long descriptions as tokens; we only reuse the precomputed KV states.

From the backbone’s perspective, multi-concept interactions are handled by attention over several disjoint KV segments instead of one, so the same decoder can perform comparisons, joint reasoning, or cross-entity references without any change in its weights.

---

### Ambiguity and practical trade-offs

Ambiguous queries such as *“What do they have in common?”* in the presence of multiple personal entities are largely a retrieval and prompt-design problem rather than a limitation of the KV interface itself:

- If the query contains explicit lexical cues (e.g., “my dog” vs “my daughter”), retrieval will typically assign high scores to the correct concepts and low scores to others, so only the relevant caches are included.
- If the query is truly ambiguous, the system can either (i) retrieve a small set of plausible concepts and let the model respond in a generic or comparative way, or (ii) ask a clarifying follow-up question. Both strategies are compatible with the current external KV mechanism.

On the scalability side, the key point is that **heavy computation remains per concept and offline**, while **per-turn work is lightweight**: retrieval plus cache concatenation followed by a single decoding pass on a short context. This is fundamentally different from approaches that must rebuild a long, multi-concept prompt and re-encode all reference images at every turn.

---

### Summary and further discussion

In summary, Jarvis’s KV-based interface extends naturally from single-concept to multi-concept personalization:

- Multi-concept queries are handled by retrieving the top few concepts and composing their external KV caches into a shared prefix.
- Complexity scales with the number of retrieved concepts, which is a controllable hyperparameter.
- The backbone and training procedure remain unchanged; all adaptation happens at the level of retrieval and cache composition.

We have added a dedicated discussion of **multi-concept personalization and cross-concept reasoning in Appendix J**, where we formalize this extension and outline potential future work on more structured routing and adaptive concept selection.

---

### Author Response · Authors · 2025-11-20
**Response to Reviewer 67Ra (W1)**

# Response to Reviewer 67Ra (W1)

---

## W1 — On Novelty, Scope, and Scalability in the Context of Personalized VLMs

> *Personalized VLMs have been extensively studied in recent years; therefore, the overall scope of this paper feels somewhat limited. I encourage the authors to propose something more novel or distinctive. The approach and design is a bit complicated, which not sure about scalability.*

**Response.**

Thank you for raising this central question. Our intent is not to add another variant of prompt concatenation or token tuning, but to change **where** personalized information lives in a VLM and **how** it is reused at inference time. Below we clarify the key ideas and how they scale.

---

### What is conceptually new

Most existing personalized VLMs follow one of two patterns:

1. **Training-required, token/adapter methods**
   (Yo’LLaVA, MC-LLaVA, RePIC, PLVM): learn concept-specific tokens or modules and inject them as extra input at every turn.

2. **Training-free, retrieval-based methods**
   (RAP-LLaVA): retrieve text or images and append them to the prompt so the backbone re-encodes all evidence each time.

Jarvis takes a different route. The core idea is to make **external, concept-specific KV states** the main personalization interface:

- For each concept, we run a one-time offline pass that encodes a compact text bundle and its mined hard patches into the frozen LLaVA-OV backbone, and we **store the resulting multi-layer KV cache as the concept’s memory**.
- At inference time, we retrieve relevant concepts and **splice their cached KV blocks directly into the current cache**, instead of re-tokenizing and re-encoding their evidence.

This leads to a distinct operating regime:

- Personalized knowledge is represented **in the model’s own KV space**, not as more tokens or extra parameters.
- Visual fingerprints from hard patches are not transient; once encoded, they become **persistent, reusable KV fragments** that can support many queries and tasks.

To our knowledge, this is the first framework that **precomputes concept-level external KV caches from both text attributes and mined visual patches, and uses them as the primary personalization mechanism for a frozen VLM**.

---

### Why the scope focuses on fine-grained QA

We agree that personalized VLMs cover a broad landscape. In this paper we deliberately focus on **fine-grained, identity-sensitive QA and recognition** for two reasons:

1. These tasks are a **stress test** for whether a personalization mechanism actually captures *who* the subject is, rather than exploiting background shortcuts or generic class cues. The Yo’LLaVA++ and MC-LLaVA++ splits emphasise micro-details and suppress easy signals by construction.

2. QA and recognition give a clean, comparable metric across methods and backbones. Under a unified LLaVA-OV backbone, Table 1 shows that Jarvis consistently improves over both training-free and training-required baselines on VQA, text-only QA, and recognition, which directly measures identity resolution rather than surface fluency.

---

### How the design scales

Although the system has several components, their cost is structured in a way that scales with the **number of concepts**, not with the **number of user turns**:

- **Offline, per-concept cost.**
  Hard-patch mining, attribute encoding, and KV construction are done once per concept. This is comparable in cost to training-required methods that learn a new token or adapter per concept, but in our case the backbone remains frozen.

- **Online, per-turn cost.**
  At inference, Jarvis performs lightweight retrieval over precomputed embeddings, loads a small number of KV blocks, concatenates them, and then runs **a single decoding pass with a short, stable context**. We never rebuild the long multi-image prompt or re-encode concept evidence. Section 4.3 and Figure 4 show that this design yields **lower latency and higher QPS** than prompt-concat and RAP-LLaVA as the number of queries per concept increases.

- **Multi-concept queries.**
  In Appendix J we describe how the same mechanism extends to multiple concepts by retrieving the top few concepts for a query and concatenating their KV caches into a shared prefix. This keeps the complexity linear in the number of retrieved concepts and does not require any modification to the backbone or additional training.

In summary, the novelty of Jarvis lies in **moving personalization into precomputed external KV space** and **explicitly mining fine-grained, concept-only hard patches that are encoded once and reused as visual fingerprints.**. The focused evaluation on fine-grained QA and recognition demonstrates that this design leads to measurable gains where identity reasoning is hardest, and the offline-per-concept / online-lightweight split ensures that the approach scales to many concepts and repeated queries.

---

### Author Response · Authors · 2025-11-20
**General Response to All Reviewers**

# General Response to All Reviewers

We sincerely thank all reviewers for the time and effort spent evaluating our submission. Your feedback was thoughtful, constructive, and highly valuable in improving both the clarity and technical depth of the paper. We appreciate the careful reading of the manuscript, the pointed questions about our design choices, and the insightful comments on scope, assumptions, and evaluation coverage.

In the revised version, we have addressed every comment in detail. We distinguish between two categories of reviewer inputs:

- **Q(i)** refers to the *i-th question* raised by reviewers, typically asking for clarification, justification, or additional explanation.

- **W(i)** refers to the *i-th weakness* identified by reviewers, usually pointing to a limitation, missing analysis, or gap in scope.

We thank the reviewers again for their constructive feedback and respond to each Q\(i\) and W\(i\) in the sections below.

---

### Author Response · Authors · 2025-11-30
**Summary from the Author**

## Dear Area Chair,

We deeply regret the unprecedented information leakage incident that has affected ICLR this year. We fully understand that the sudden surge in workload and the resulting disruption must be incredibly challenging for you and the other organizers.

As authors, we wish to provide a concise summary of the **"Initial Review"** and our **"Rebuttal Updates"** to assist in your decision-making process.

## 1. Initial Review Highlights

We are encouraged by the **positive reception** and the **consistent recognition** of our work across two key dimensions:

### **• Method & Practicality:**
- Described as *"novel and practical"* (*Reviewer JB1P*)
- Acknowledged as *"practical and easy to deploy"* (*Reviewer tLpQ*)
- *"I like the idea of how they reduced latency"* (*Reviewer 67Ra*)

### **• Performance & Efficiency:**
- Praised for *"superior performance"* (*Reviewer DAdL*)
- Noted that it *"significantly reduces latency and improves throughput"* (*Reviewer 67Ra*)
- *"Provides strong evidence of practical benefits"* (*Reviewer JB1P*)
- *"Effectively reduce computational overhead during inference"* (*Reviewer tLpQ*)

## 2. Rebuttal Clarifications

Due to space constraints in the initial submission, the motivation behind certain designs such as the **hard patch mining pipeline** caused some confusion. We have clarified this extensively in our response.

**To illustrate the depth of our rebuttal:**
> We clarified that our algorithm captures the subject’s unique "visual fingerprint" by identifying regions that are simultaneously *difficult to synthesize* and *semantically relevant*. Specifically, we leverage the unconditional diffusion prior to detect "hard-to-generate" details (via the difficulty map $C_m$) and fuse this with text relevance ($R_m$) to filter out complex but irrelevant backgrounds. By further enforcing subject coverage and using region averaging, we ensure the External KV robustly encodes the target concept's identity.

We have provided similarly detailed explanations for all other concerns in our **point-by-point responses** and have incorporated these improvements into the **revised PDF**. Please feel free to review them thoroughly.

## Conclusion

Regrettably, due to the incident, we were unable to engage in the discussion phase, and reviewers have not yet responded to our rebuttal. Nevertheless, we are confident that the technical solidity of our manuscript, combined with our detailed clarifications, effectively addresses the initial concerns.

We trust in your professionalism and your ability to make an objective decision despite these emergency circumstances. Thank you for your dedication and hard work.

Sincerely,

**Authors of Submission 8278**

---

### Meta-Review · Area_Chair_RGzU · 2026-01-06

**Summary:**

Reviewers generally acknowledged this paper's **practical motivation** and reported **efficiency gains**, but the decision-relevant concerns were substantial:

1. The method was initially **hard to follow**, with unclear definitions and insufficient intuition for the hard-patch mining pipeline. Authors add more explanations in the revision during rebuttal, while the whole system still looks complicated and hard to follow.

2. **Novelty remained uncertain** compared to existing retrieval- and KV-cache-based personalization. Authors emphasize that their contributions lie in externalizing concept kv-cache for personalization, but it still doesn't look quite convincing.

3. **Narrow and potentially insufficiently convincing evaluation**: this paper focuses largely on single-concept QA with limited direct evidence for multi-concept personalization, dynamic updates, and generalization to more complex personalized reasoning.

While the rebuttal improved explanations and added some ablations/baselines, key concerns about scope and the strength of empirical validation for real-world personalization settings remain unaddressed. Thus, I recommend Reject.

**Reviewer Concerns:**

Some concerns about the paper presentations, additional baseline comparisons and hyperparameter ablations are basically addressed.

However, there are still a few concerns remain unsolved or partially solved:

1. **Generalizing to multi-concept setting** raised by **Reviewer JB1P, DAdL, tLpQ**: The authors only provide a discussion about how to generalize their method to handle multi-concepts. However, no qualitative and quantitative comparison with existing methods for this scenario, leaving this concern partially addressed. The authors also recognize their method is limited to generalize to multi-concept setting with high efficiency, making it difficult to deal with complex personalized queries.

2. **Novelty beyond prior work** raised by **Reviewer 67Ra, tLpQ**: Although the rebuttal argues “concept-indexed external KV + hard-patch fingerprints” is distinct, the difference still feels incremental to retrieval-based personalization and KV-cache reuse.

3. **Overall presentation** raised by **Reviewer DAdL**: While authors put a lot of efforts in updating appendix, concerns may remain if the main paper still relies heavily on appendix material for core definitions and motivation.

**Reviewer Scores:**

For **Reviewer DAdL (score 2)**, he probably stays at 2. Although many concerns about the method are clarified, he may still consider the core evaluation scope as insufficient.

For **Reviewer 67Ra (score 6)**, he probably drops to 4 as his concerns about the novelty and the scope are only partially alleviated. The scalability issue raised by him is also not fully addressed with qualitative results.

For **Reviewer JB1P (score 4)**, he probably stays at 4. Although the conceptual multi-concept and update mechanisms are clarified, the lack of empirical validation may prevent a score increase.

For **Reviewer tLpQ (score 4)**, he probably also stays at 4. His concerns about novelty, broader task coverage, and real-world multi-concept generalization remain partially resolved, which may keep his final score below accept.

---

### Decision · Program_Chairs · 2026-01-26

Reject